



**Validation of MODIS 3 km Land Aerosol Optical Depth from NASA's EOS Terra and**
**Aqua Missions**
Pawan Gupta[1,2], Lorraine A. Remer[3], Robert C. Levy[2], Shana Mattoo[2,4]
[1] {Goddard Earth Sciences Technology And Research (GESTAR), Universities Space
Research Association (USRA), Columbia, MD, USA}
[2] {NASA Goddard Space Flight Center, Greenbelt, MD 20771, USA}
[3] {JCET, University of Maryland – Baltimore County, Baltimore, MD 21228, USA}
[4] {Science Systems and Applications, Inc, Lanham, MD 20709, USA}
Correspondence to: Pawan Gupta (pawan.gupta@nasa.gov)
**Abstract**
The two MODerate Resolution Imaging Spectroradiometer (MODIS) sensors, aboard Earth
Observing Satellites (EOS) Terra and Aqua, have been making aerosol observations for more
than 15 years. From these observations, the MODIS dark target (DT) aerosol retrieval algorithm
provides aerosol optical depth (AOD) products, globally over both land and ocean. In addition to
the standard resolution product (10x10 km$^2$), the MODIS collection 6 (C006) data release
included a higher resolution (3x3 km$^2$). Other than accommodations for the two different
resolutions, the 10 km and 3 km DT algorithms are basically the same. In this study, we perform
global validation of the higher resolution AOD over global land by comparing against
AERONET measurements. The MODIS-AERONET collocated data sets consist of 161,410
high-confidence AOD pairs from 2000 to 2015 for MODIS Terra and 2003 to 2015 for MODIS-
Aqua. We find that 62..5% and 68.4 % of AODs retrieved from MODIS-Terra and MODIS-
Aqua, respectively, fall within previously published expected error bounds of ±
(0.05+0.2*AOD), with a high correlation (R=0.87). The scatter is not random, but exhibits a
mean positive bias of ~0.06 for Terra and ~0.03 for Aqua. These biases for the 3 km product are
approximately 0.03 larger than the biases found in similar validations of the 10 km product. The
validation results for the 3 km product did not have a relationship to aerosol loading (i.e. true
AOD), but did exhibit dependence on quality flags, region, viewing geometry, and aerosol





spatial variability. Time series of global MODIS-AERONET differences show that validation is not static, but has changed over the course of both sensors' lifetimes, with MODIS-Terra showing more change over time. The likely cause of the change of validation over time is sensor degradation, but changes in the distribution of AERONET stations and differences in the global aerosol system itself could be contributing to the temporal variability of validation.

## 1. Introduction

The Moderate Resolution Imaging Spectroradiometer (MODIS) sensors, onboard the Earth Observing System (EOS) Terra and Aqua satellites, have been providing observations of Earth and the atmosphere for almost two decades (Salomonson et al., 1989). These data have been used to create a long-term set of atmospheric aerosol properties including aerosol optical depth (AOD – a measure of aerosol loading in the total atmospheric column) (Kaufman et al., 1997; Levy et al., 2013). In particular we are addressing products of the Dark Target (DT) algorithms that provide aerosol retrievals over both ocean and dark vegetated land surfaces (Kaufman et al., 1997, Remer et al., 2005; Levy et al., 2007a; 2007b; 2013). The DT products were designed with climate applications in mind and have been used to address a wide variety of geophysical science questions including the role of aerosols in climate-relevant processes (Kaufman et al., 2002; Christopher et al., 2002; Yu et al., 2006), cloud/precipitation modifications (Koren et al., 2009; 2012; Yuan et al., 2011; Oreopoulos et al., 2016), and long-range transport of aerosols (Kaufman et al., 2005; Yu et al., 2012). Users have even applied the DT aerosol product to address needs for monitoring, evaluating and forecasting air quality (al Saadi et al., 2005; Gupta et al., 2009; Van Donkelaar et al., 2015).

The MODIS DT algorithm produces an aerosol product, over land and ocean, at a nominal 10 x10 km$^2$ spatial resolution. This spatial resolution permits much selectivity in choosing which MODIS-measured reflectance pixels at 0.5 x 0.5 km$^2$ resolution to include in the retrieval, and generally produces smooth and accurate fields of AOD and other aerosol parameters (Remer et al., 2012). By allowing the algorithm to discard up to 90% of the available pixels and still produce a high quality aerosol product, the algorithm avoids marginal situations unfavorable for an aerosol retrieval such as cloud fringes, fragments and shadows, as well as land surfaces that





do not agree with algorithm assumptions (Remer et al., 2012).    The 10 km product has
undergone lengthy evaluation and validation, updated after each major algorithm modification
(Ichoku et al., 2002; Chu et al., 2002; Remer et al., 2002, 2005; Russell et al., 2007; Levy et al.,
2005; 2010).  Some of this evaluation was global in nature, while some local to a particular field
experiment, but all concerned the 10 km MODIS DT aerosol product.
For climate studies, the initial intention of the algorithm, 10 km spatial resolution was sufficient
to characterize global and regional aerosol loading.  However, as the community expanded the
use of MODIS AOD to a wide variety of purposes, need arose for a finer resolution product, and
a nominal 3 x 3 km$^2$ resolution product was introduced as part of MODIS Collection 6 (Levy et
al., 2013; Remer et al., 2013).  The product is termed MYD04_3K for 3 km resolution aerosol
parameters derived from the MODIS-Aqua sensor and MOD04_3K for those derived from
MODIS-Terra.  These products are produced operationally, over land and ocean, and the entire
data records of Terra and Aqua have been reprocessed, creating a data record of almost two
decades.
Before becoming operational, Remer et al., (2013) tested the algorithm by comparing six months
of global 3 km retrievals from MODIS-Aqua against available ground truth, while other
independent studies (Munchak et al., 2013; Nichol and Bilal, 2016; He et al., 2017 and others)
have done subsequent evaluation of the product regionally and locally.   These limited
comparisons suggested that the new AOD product would be sufficiently accurate to provide
useful information and new perspective to the aerosol community, but might introduce additional
noise and/or bias that the original coarser resolution product successfully avoided. Now that the
multi-decadal 3 km product is operational and available publicly, it is time to perform a
comprehensive evaluation of this finer resolution MODIS DT aerosol product.  We present here
the results of an analysis of a comparison of the global long-term MODIS 3 km product with
collocated AErosol RObotoic NETwork (AERONET) (Holben et al., 1998) observations.
**2.  The MODIS dark target 3 km aerosol retrieval over land**



The MODIS DT algorithm and products are described in detail in Levy et al., (2013) and also in
the MODIS DT on-line Algorithm Theoretical Basis Document (ATBD, 2017). In summary, to
retrieve aerosol parameters over land, the algorithm makes use of the reflectances measured in
three of MODIS' 36 spectral channels, 0.47 µm, 0.65 µm and 2.1 µm (Levy et al., 2007a).  These
are provided in nominal spatial resolution of 0.5 x 0.5 km$^2$ (at nadir) other channels (some at 0.5
x 0.5 km$^2$, some at 1 x 1 km$^2$ resolution) are used for identifying appropriate surfaces for
retrieval, and masking clouds, snow and ice.  While the 10 km$^2$ (standard product – MxD04_L2,
x=O for Terra, x=Y for Aqua) begins with 20x20 = 400 aggregation of 0.5 km$^2$ native-resolution
pixels, the 3 km$^2$ aerosol retrieval box starts with 6 x 6 = 36 aggregation of such pixels.  Native
pixels are removed, as to retain only the ones most appropriate for a dark-target, over-land
retrieval. Native pixels tagged as too bright for the dark target algorithm, or identified as
containing cloud, or a water, snow or ice surface are removed from the aggregation.   The
remaining native pixels are sorted from darkest to brightest, and the darkest 20% and brightest
50% of all remaining native pixels are removed as well. For the 3 km$^2$ retrieval this means at
most 12 native pixels will remain, and likely fewer.  For minimum statistical robustness, the 3km
algorithm requires at least 5 native pixels (out of the initial 36).  If there are insufficient native
pixels (e.g <5), output parameters are assigned fill values and no retrieval is attempted.  Based on
the aggregation and filtering, it is likely that there will be pixels native used by the 3 km$^2$
retrieval that would have been discarded by the coarser (10 km$^2$) standard resolution product
(Remer et al., 2013).   For whichever product resolution, the remaining native pixels are
averaged, leading to a single set of spectral reflectance values that drives the aerosol retrieval.
Except for downstream decisions based on number of native pixels used, the 10 km$^2$ and 3 km$^2$
retrievals proceed identically.
The retrieval uses a Look Up Table (LUT) procedure in which the LUT is constructed a priori of
simulated top-of-atmosphere reflectances, calculated using assumptions of aerosol optical
properties based on AERONET inversions (Dubovik and King, 2000) and radiative transfer. The
surface reflectance is constrained by assuming a relationship between reflectance at 2.1 µm with
the reflectances at 0.65 µm and 0.47 µm that is a function of a Normalized Difference Vegetation
Index using Short Wave Infrared (SWIR) bands (NDVI$_{SWIR}$) and scattering angle (Levy et al.,
2007a, 2007b; 2013).  The algorithm finds the AOD that minimizes the differences between the



MODIS-observed mean spectral reflectances of the retrieval square and the simulated reflectance
values of the LUT. The primary output of the retrieval is the AOD at 0.55 μm. Using a series of
tests the algorithm assigns a quality assurance flag (QAF) of either 3, 'good quality' or 0, 'bad
quality' to the retrieval. These values can be interpreted as "confidence" in the aerosol retrieval
(whether the retrieval proceeded nominally, and whether there are enough native-resolution
pixels). For quantitative use of the 10 $km^2$ product, the MODIS DT team has recommended
limiting to QAF=3 retrievals (≥51 native pixels out of 120). The similar ratio for the $3km^2$
product is ≥5 out of 12. Any fewer, and there is too little statistical information for any
confidence in an aerosol retrieval. Therefore, for the 3 $km^2$ product, any fewer than 5 native
pixels automatically receives QAF=0. QAF values assigned as 1 or 2 are based on other criteria.
**3. Data Sets**
**3.1. MODIS 3 km AOD**
The primary data set of this study is the Collection 6 MODIS dark target retrieved aerosol optical
depth at 3 km spatial resolution, derived from Terra reflectances (MOD04_3K), or Aqua
reflectances (MYD04_3K), as described in Section 2. These are publicly available and can be
downloaded from https://ladsweb.modaps.eosdis.nasa.gov/. Of the products in the data sets, we
analyze only the AOD at 0.55 μm.
Applying identical algorithms to two different sensors does not guarantee identical results (Levy
et al., 2015). The two MODIS DT data sets, one from MODIS-Terra and one from MODIS-
Aqua must be addressed separately as individual and independent products, even though they
have been created from identical algorithms with no specific tuning of parameters for each
sensor. While MODIS-Terra and MODIS-Aqua began as near-identical sensors, they have
evolved over their lifetimes to develop their own instrumental characteristics. For example,
some detectors in Aqua's detector array at some wavelengths have died, resulting in fewer
available reflectance pixels at those wavelengths. Terra's detector array has not lost any
detectors. At the same time, we have seen drift in some of Terra's wavelengths, resulting in
measureable artificial trends in MODIS-Terra' aerosol products (Levy et al., 2013; Sayer et al.,





2015; Lyapustin et al., 2014). The most flagrant of those MODIS-Terra trends have been
mitigated by aggressive radiometric calibration (Toller et al., 2013), which has been applied for
creating the C006 DT products. Note that some projects (e.g. Lyapustin et al., 2014; Sayer et al.,
2015) have since introduced additional mitigation, however, the DT retrieval has not applied
these strategies. In this work, we will separately analyze the C006 aerosol products from the two
MODIS sensors, (MOD04_3K and MYD04_3K), to provide users with clear information on the
strengths and limitations of each one.

## 3.2.    AERONET AOD

The Aerosol Robotic Network (AERONET) is NASA's global ground network of CIMEL sun-
sky radiometers that make measurement of direct transmitted solar light and scattered sky light at
several wavelengths during daylight hours (Holben et al., 1998). In this work, only the direct sun
measurements will be used. AERONET processes these spectral measurements to derive AOD in
their respective wavelengths. The AERONET spectral AOD product is a community standard for
satellite-derived AOD validation, given that AERONET's AOD uncertainty of 0.01-0.02 (Eck et
al., 1999) is sufficiently more accurate and precise than can be expected by any satellite retrieval.
The typical temporal frequency of direct sun measurements is every 15 minutes.  The network
consists of hundreds of stations, located globally, across all continents and in a wide variety of
aerosol, meteorological and surface type conditions. Only stations that sufficiently represent land
areas will be used here, which means we are not comparing with observations taken on small
islands, ocean platforms or mobile ships. The validation procedure requires calculating the
spatio-temporal statistics of a collocated MODIS-retrieved and AERONET-measured AOD pair
(Ichoku et al. 2002; Petrenko et al., 2012; Munchak et al., 2013; Remer et al., 2012).  Figure 1
shows the location these stations and color-coding represents the number of collocated
AERONET-MODIS AOD pairs over the station.

The configuration of the spectral bands varies, but typically is centered at 0.34, 0.38, 0.44, 0.50,
0.67, 0.87, and 1.02 µm. Here we use a quadratic log-log fit (Eck et al., 1999) to interpolate
AERONET AOD to 0.55 µm to match the primary MODIS AOD product. AOD data from
AERONET are reported for three different quality levels: unscreened (level 1.0), cloud screened



(level 1.5) and cloud screened and quality assured (level 2.0). We will only use level 2.0, version
2.0 AERONET AODs in this study.
**4.   Spatial and Temporal Collocation**
We have created a collocated data set (CDS) of both MODIS-Terra and MODIS-Aqua with
AERONET for nearly the entire mission (2003-2015 for Aqua and 2000-2015 for Terra).
MODIS Collection 6, 3km AOD retrievals over land are collocated with AERONET version 2.0,
level 2.0 AOD measurements using a modified spatio-temporal technique as outlined by (Ichoku
et al., 2002; Petrenko et al., 2012; Munchak et al., 2013; Remer et al., 2013). From here on, we
use the term "pixels" to refer to the MODIS retrieval product (e.g. 3 or 10 $km^2$ resolution); if
referring to the native MODIS pixel resolution (e.g. 0.5 $km^2$) we will denote as "native pixel".
In previous validation studies of the standard 10 km product the spatio-temporal statistics were
based on either 5 x 5 MODIS product pixels (~50x50 $km^2$ box) centered on the AERONET
station (Ichoku et al., 2002; Levy et al., 2010) or all the MODIS product pixels within a 27.5 km
radius around the AERONET station (Petrenko et al., 2012) matched with the temporal statistics
of ±30 minutes of AERONET observations centered at satellite overpass time.   These large
spatial collocation boxes will not properly test the accuracy of finer resolution satellite products
to represent small-scale aerosol gradients.  Therefore, Remer et al., (2013) and Munchak et al.
(2013) moved to a 7.5 km radius and ±30 minutes of overpass.   The 7.5 km radius encompasses
roughly 25 AOD pixels at nadir, which is analogous to the number of product pixels used with
the coarser resolution product. In this study spatial statistics are calculated from all MODIS
product pixels falling within a box of 0.15°x0.15° (latitude x longitude) centered over an
AERONET location. Except for Polar Regions, this is similar to a 15 x 15 $km^2$ box or 7.5 km
radius search at the nadir. Temporal statistics are calculated from all AERONET observations of
AOD within ±30 minutes of satellite overpass.
As recommended by the MODIS DT science team (Levy et al., 2010), unless otherwise
specified, only AOD pixels with quality assurance flag 'very good' (QAF=3) were included in
averaging over the AERONET sites. To be consistent with previous validation exercises (Levy et





al., 2010), we have retained the collocated data sets only when there were at least 5 MODIS
product pixels (out of a possible 25) and 2 AERONET measurements (out of a possible 2-4). The
collocated data set (CDS) consists of 574 AERONET stations with 90,162 collocated pairs for
MODIS-Terra and 71,248 collocated pairs for MODIS-Aqua. Figure 1 shows the number of
CDS pairs at each AERONET station, for Terra and Aqua.
Thus, a data set (i.e. CDS) of collocated MODIS-AERONET pairs of AOD at 0.55 μm is created
that can be organized and subsampled in any number of configurations.  In any subsample, or for
the entire data set, these ordered pairs can be plotted, one against the other to create a scatterplot,
and collocation statistics calculated.  We will use the following statistical parameters to quantify
how well the MODIS retrievals match their collocated AERONET counterparts (Hyer et al,

11   2011):

• Correlation coefficient (R),
• Slope of the linear regression line,
• Root Mean Square Error (RMSE)

$$\text{Mean Bias} = \frac{1}{N}\sum(\text{MODIS AOD} - \text{AERONET AOD}) \cdots\cdots\cdots\cdots (1)$$

Percentage of collocations falling within expected error,

$$\text{EE} = \pm(0.05 + 0.20\times\text{AERONET AOD}) \cdots\cdots\cdots\cdots\cdots\cdots (2)$$

Error Ratio (ER),

$$ER = (\text{MODIS AOD} - \text{AERONET AOD})/\text{EE} \cdots\cdots\cdots\cdots\cdots\cdots (3)$$

The coefficients in the EE equation were determined from evaluation of the 3 km$^2$ product over
the six months of Aqua data analyzed by Remer et al., (2013). Those limited results suggested
that expected error bounds should be broadened to the values seen in Eq. (2) from those derived
for the 10 km product (EE=±0.05±0.15 x AERONET AOD).
The number of collocations (N) is another parameter used to evaluate the 3 km$^2$ retrieval in the
collocation data set.




**5.  Validation Results**
**5.1. Global Statistics**
We first compare MODIS 3 km$^2$ AOD retrievals against collocated AERONET values, for both
the recommended 'high quality' retrievals (QAF=3) and for all the retrievals, regardless of
quality, keeping Terra and Aqua results separate.  Results are plotted in Figure 2.  Note that the 3
km$^2$ product only tags data as either 'high quality' or 'low quality'. Table 1 presents the
statistical parameters corresponding to this analysis while considering various combinations of
QAFs.
Globally, there is strong correlation between MODIS 3 km$^2$ AOD and collocated AERONET
equivalents.  However, there is scatter and a positive bias to the retrievals, more so for Terra than
Aqua, even though the correlation is similar between the satellites. Identified retrieval quality
matters to product accuracy with QAF=3 showing stronger correlation, smaller RMSE and more
retrievals falling within expected error than QAF=0, but the high quality data set loses about
20% of the retrievals. Figure 3 shows that the differences between Terra and Aqua in how they
match AERONET values are much more apparent than the differences between QAF levels of
the same satellite sensors. We note that only the high quality (QAF=3) Aqua 3 km$^2$ retrievals
meet expectations in terms of falling within the standard expected error bars (Remer et al., 2012;
and Eq. 2).
Table 1 also shows the corresponding validation statistics for the 10 km$^2$ product for QAF=3,
distinguishing between Terra and Aqua.  The 10 km$^2$ product, as expected, more closely matches
AERONET values, having higher correlation, lower bias and RMSE, and producing more
retrievals that fall within expected error bounds than does the 3 km$^2$ product.  We note that even
in the 10 km$^2$ validation statistics, mean bias for Terra is 0.03 higher than for Aqua, which is the
same difference between sensors as found for the 3 km$^2$ product. The results in Table 1 confirm
Remer et al. (2013)'s conclusion that the 3 km$^2$ product is less accurate than the standard 10 km$^2$
product.  The remainder of the paper will be devoted to exclusively analyzing the differences





between the 3 km$^2$ product and AERONET, without further reference to the standard 10 km$^2$
product.
**5.2. Regional Statistics**
The accuracy of the 3 km AOD retrievals will be regionally and locally specific, depending on
how well retrieval assumptions of surface and aerosol optical properties match actual conditions,
local cloud conditions that may or may not introduce uncertainty into the retrieval and the
spatial/temporal variability of the area that may create biases in the collocation methodology that
depends on assumptions of aerosol homogeneity. Here we investigate how well the MODIS 3
km$^2$ product matches AERONET over individual AERONET stations.
For the regional and local analyses, we will use only QAF=3 retrievals and calculate the same
collocation statistics for each station, individually. Figure 4 plots the values for correlation
coefficient, mean bias, percentage within expected error, and RMSE for each station that
reported at least 100 collocations over the entire time series. In general, the MODIS 3 km$^2$
retrievals show high correlations over much of the northern mid-latitudes where there are
AERONET stations in abundance. This relative abundance drives the global statistics of Table
1. Correlation breaks down at some stations in California and the arid southwest of North
America, in the Caribbean and Central America, Insular SE Asia and Australia, and especially in
southern South America.
Not all stations with strong correlations exhibit small mean biases. For example, MODIS 3 km$^2$
retrievals severely under predict AOD in the stations of west Africa, falling well below expected
error there, even though those stations report high correlations with AERONET.  Stations in
Australia, show relatively small mean biases and high percentages meeting expectations, despite
poor correlations, suggesting that the poor correlations are the result of small dynamic range in
the scatter plots that occur when AOD is consistently low.
In Figure 4, we see the local nature of the validation statistics. Stations in close proximity to
each other can report very different statistics. For example, the stations clustered across northern





India, and those in an array across central South America (Brazil) range from strong positive to
negative mean biases and RMSE error from 0.05 to 0.20, even though these groupings of stations
will fall within the same region as defined in Figure 5. This is apparent in almost any region.
Some of this variability may be due to differences in the temporal extent of the AERONET
record at each individual station, so that even if stations are in close proximity in space, they may
actually be making measurements in entirely different years or seasons. Other differences may
be related to topography, urban surfaces, or other factors. Still, the variability seen in Figure 4
shows how local conditions, and possibly the individual characteristics of the time series affect
validation statistics.
The final point to note in Figure 4 is the difference between Terra and Aqua. For example, in the
mean bias plots we see how the mean bias across the North American central plains fades from
approximately positive 0.04-0.05 to slightly negative. For many of the stations, positive mean
biases decrease from Terra to Aqua. This is in agreement with the global statistics presented in
Table 1.
For the regional analysis, we will use only QAF=3 retrievals and calculate the same collocation
statistics for all stations within each region. The 17 regions were defined following Hyer et al.,
(2011), and as shown in Figure 5
Table 2 presents the regional validation statistics for each region defined in Figure 5. We know
from previous analyses presented above that there are distinctive differences between Terra and
Aqua mean biases; however, in calculating the regional statistics of Table 2, we combine Terra
and Aqua collocations. Only QAF=3 retrievals are included.
The majority of collocations are found in the northern mid-latitude regions, with E. and W.
CONUS (East and West Continental United States) representing 25% of the total collocations
and Europe Mediterranean and Eurasian Boreal representing another 34% of the total. MODIS 3
km retrievals from E. CONUS, Europe Mediterranean and Eurasian Boreal show very good
overall agreement with AERONET, exhibiting R≥0.78, bias ≤ 0.05 and at least 2/3 of retrievals
falling within expected error. W. CONUS retrievals agree with AERONET less well, exhibiting





some of the highest positive biases of any region on the globe. These four regions drive the
global validation statistics, which reflect both the good agreement of E. CONUS and Europe, and
the high bias of W. CONUS.
Regions where MODIS 3 $km^2$ retrievals exhibit especially good agreement with AERONET
collocations include E. CONUS and Europe, as mentioned above, as well as north/central South
America, equatorial and southern Africa, and Australia. Australia is particularly interesting,
because even though its correlation is low, 75% of retrievals there fall within expected error
bounds, suggesting small dynamic range in the scatter plots. Regions where retrievals exhibit
especially poor agreement with AERONET include W. CONUS, as mentioned above, plus
Central America, SW Asia, East Asia Mid-Latitudes, south South America and Insular Southeast
Asia. East Asia is interesting because of its high correlation, but with insufficient retrievals
falling within expected error bounds. It appears that the MODIS retrieval is sensitive to aerosol
there, but is incorrectly modeling the aerosol and surface optical properties. Validation statistics
are especially poor in Southwest Asia, where there are very few stations and collocations.
**5.3 Error dependencies**
We next explore the relationship between MODIS-AERONET 3 $km^2$ AOD differences and
various parameters for the global collocation data set, keeping Terra and Aqua collocations
separate. At each collocation, the AERONET AOD is subtracted from the MODIS AOD, so that
a positive difference indicates a positive MODIS bias. The data is then sorted according a
particular parameter in the database. Collocations are grouped into 87 bins for Terra and 67 bins
for Aqua, each containing 1000 collocations. Thus, there are equal numbers of collocations in
each bin, but the bins are not equally spaced along the x-axis, this way to calculated statistics
free from the sample size. The mean, median and standard deviations of the MODIS-AERONET
differences are calculated for each bin.
Figure 6 shows the results of this analysis as a function of AOD, both the true AOD, as measured
by AERONET and by the MODIS-retrieved AOD. The AOD differences depend very little on
the true AOD. There is some suggestion of a positive-negative-positive shift of differences at the



very lowest AOD (<0.1), but overall the differences are flat. Terra exhibits an overall positive
mean bias against AERONET of about 0.06, with the bias in Aqua much less noticeable.
Plotting these differences against the MODIS-retrieved AOD provides the user to evaluate an
individual retrieval, given only a retrieved parameter. Here we see a distinctive pattern between
MODIS AOD bias and MODIS AOD. The higher the retrieved AOD, the greater the positive
difference between MODIS and AERONET. Significant biases of >0.10 are seen for MODIS
AOD values > 0.40. For retrieved AOD < 0.10, the mean differences between MODIS and
AERONET are negative. This indicates that a high value of retrieved AOD has greater
probability of being too high than too low, and a low value of retrieved AOD has a greater
probability of being too low than too high. These results are expected, as high AOD retrievals
are more sensitive to true aerosol properties whereas true surface properties become more
important in low AOD retrieval.
Figure 7 shows how MODIS-AERONET AOD differences vary as function of AOD variability
and availability of high-resolution (0.5 km$^2$) native pixels for retrieval to be performed at 3 km$^2$
resolution. Standard deviation of the retrievals in the 5x5 collocation box is a measure of the
homogeneity of the aerosol across the box. The collocation methodology assumes that MODIS
spatial statistics will match AERONET temporal statistics, which holds best if the aerosol field is
homogeneous in space and time at the AERONET station. As variability across the box
increases (i.e. STD (AOD)), we expect differences between MODIS AOD and AERONET AOD
to grow. However, we do not have a priori expectations of whether that growth will be positive
or negative. We see from Figure 7 (top row), that differences are increasingly positive as
variability increases. This is because the standard deviation is not normalized, and the
differences increase simply because the AOD is increasing as it does in Figure 6.
Another test of the collocation methodology assumptions is to look for error dependencies on the
number of MODIS retrievals within the 5x5 collocation box. Note that the methodology requires
at least 5 retrievals to represent the box and may have as many as 25. We see from Figure 7
(middle row) that there are dependencies. Fewer numbers of retrievals are associated with
positive differences, but having almost all of the 25 retrievals available is associated with
negative differences. We understand, how collocation statistics might be skewed by having



fewer numbers of retrievals available to match AERONET, especially if the aerosols across the
collocation box were not spatially homogeneous. Also fewer numbers of retrievals may be a
result of marginal retrieval conditions caused by clouds and unfavorable surface conditions. It is
less easy to understand the negative differences when the box is especially well represented with
sufficient retrievals, and this require further investigation on individual retrieval.
The bottom row of Figure 7 shows the MODIS-AERONET AOD differences as a function of the
average number of reflectance native pixels ($0.5 \text{ km}^2$) used by the MODIS $3 \text{ km}^2$ retrieval in
producing a value of AOD. The retrieval begins with 36 native pixels, and after masking, sorting
and discarding; we left with between 5 and 12 pixels. The number of pixels used by the retrieval
is indication of how much masking was required. If 12 pixels remain, then no masking was
required and the situation is cloud-free and over favorable surfaces. If only 5 pixels remain,
there are conditions in that retrieval that could raise concerns. In Figure 7, we see that the fewer
the pixels used by the retrieval (i.e., more masking is needed), the higher the positive bias,
especially for Terra. This suggests, in the Terra retrieval, that clouds or unfavorable surface
conditions are contributing to the high bias we are seeing in the global data set. Interestingly,
MODIS-AERONET differences are negative when masking is at a minimum, similar as to when
the collocation box contains almost all possible retrievals. It seems that cloud-free situations
with appropriate surface features are associated with MODIS under predicting AERONET AOD.
The same functional relationship is apparent in the Aqua data set also, but the biases, both high
and low, are less pronounced.
Figure 8 shows the MODIS – AERONET AOD differences as a function of geometry. The top
row plots the differences against scattering angle, where we see positive bias increasing towards
the extreme backscattering angles. The functional relationship is similar in both Terra and Aqua,
but Terra's positive bias is more pronounced. The bottom row plots the differences against
sensor view (i.e. zenith) angle, where the Aqua differences show little dependence on view
angle, but the Terra differences increase positive biases in near nadir views. Geometrical
dependencies in bias generally point to systematic inaccuracies in retrieval assumptions. These
can be either in terms of surface angular functions or in aerosol optical properties. However, the
difference between Terra and Aqua sensor zenith angle dependencies suggests an issue with



instrument characterization, which could include geometrical functionality due to the need to
calibrate across the scan mirror.
**5.4 Temporal Changes**
Examining temporal changes of validation statistics across the entire time series of the
collocation database further characterizes the accuracy of the 3 km$^2$ AOD product.  Figure 9
plots monthly mean error ratios (Eq.3), and number of collocations for the time series of Terra
(red) and Aqua (blue), separately. The error ratios (ER) compare the actual error (bias) to the
expected errors (EE). The -1≤ER≤1 means the actual errors are smaller than EE whereas |EE|>1,
indicates a poor match.   Even if the MODIS sensors and the algorithm were entirely consistent
during the time series, AERONET stations go on and off line, causing global validation statistics
to shift in local and regional emphasis, and introducing temporal variation in the global results.
Therefore, we have selected 26 AERONET stations (Table 3, Figure 9) with long-term data
records with consistent collocation over the entire time series for this analysis. The analysis over
these selected stations allow us to examine the change in bias (and error ratios) over a longer
time period without change in spatial and temporal distribution of AERONET stations. Only
QAF=3 retrievals are used. During the 15 years of the collocation data set many factors have
changed.  For example, satellite sensor characterization is an ongoing process that employs
several different measures of radiometric drift and then continuously adjusts calibration
parameters to compensate for that drift.  Thus, even though the algorithm remains consistent
throughout the data record, the inputs to that algorithm may not be, despite the best efforts of the
MODIS Characterization Team.
The time series of the monthly statistics shows strong seasonal variation of mean bias and
number of collocations.  Strong positive bias occurs in the April-August time period, followed by
low or even negative bias in the October – February period
In addition to the seasonal variability, Figure 9 also exposes long-term temporal trends.  There is
a steady increase of the number of collocations per month, nearly doubling from the early years,
up and through the beginning of 2012, as the AERONET network expands over time. The last





few years of the record show a decrease in collocations, in some part attributed to the lag in promoting AERONET records from Level 1.5 to Level 2.0. We only use AERONET Level 2.0 for collocations.

The temporal mean biases for the entire time series are 0.04 and 0.014 for Terra and Aqua, respectively, corresponding to Error Ratio (ER) in Figure 9 of 0.55 and 0.2. The mean biases also exhibit temporal trends with biases beginning to increase around 2008/2009. The bias for Terra increased from 0.038 to 0.048 whereas these numbers for Aqua are 0.014 and 0.016. The corresponding ER increase for Terra in 2008 is from 0.48 to 0.65. The increase in ER for Aqua is negligible.

The systematic higher biases exhibited by Terra as compared with Aqua agree with the global analysis presented above. This offset in bias between the two MODIS sensors appears systematic from the beginning of the Aqua record to the end of the time series, although the magnitude of that offset increases over time as Terra's biases grow. The systematic greater number of collocations in the Terra data set than in Aqua's may result from diurnal cloud patterns that create cloudier conditions in the afternoon during Aqua overpass than during Terra's morning one. More clouds in the afternoon (King et al., 2013) may reduce the number of possible collocations. However, instrumental differences affecting available retrievals are another possibility.

## 6. Discussion and Conclusions

To validate the MODIS 3 km$^2$ AOD products (MOD04_3K and MYD04_3K), which became publicly available in the MODIS Collection 6 release, we created a database of collocations between the product and AERONET observations for the extent of the MODIS record from 2000 – 2015. Collocation criteria employed 5x5 MODIS retrievals centered at the AERONET station and all AERONET observations ±30 minutes of satellite overpass. Thus, the collocation box is approximately 15 km per side, for nadir views. Only Level 2, quality assured AERONET observations are included, and AERONET AOD is interpolated to 0.55 μm to match MODIS



values.  Overall there are over 90,162 high quality collocations of Terra retrievals and over
71,248 high quality collocations for Aqua.
The validation statistics examined include mean bias, regression slope, correlation coefficient
and percentage falling within expected error bounds.  In this validation exercise we hold the 3
km$^2$ AOD product to expected error thresholds of ±0.05±20% (Remer et al., 2013). We find that
the global 3 km$^2$ AOD product displays skill in matching AERONET observations with a
correlation coefficient of 0.87, but there is RMSE of 0.15 and 0.13 for Terra and Aqua,
respectively.  The scatter is not random, but exhibits a mean positive bias of 0.06 for Terra and
0.03 for Aqua. The coarse product error bounds capture 2/3 of the Aqua 3 km AOD retrieval, but
less than 63% of the Terra retrievals.  There is significant degradation of validation accuracy if
MODIS retrievals of Poor data quality (QA<3) are included in the analysis.  Approximately 20%
of all MODIS retrievals were identified as Poor quality. Thus, on a global basis we recommend
using only QAF=3 MODIS 3 km$^2$ retrievals for quantitative analysis.  If doing so, then the
expected error for the Aqua product is ±0.05 ± 0.20AOD, on a global basis, but only ±0.06 ±
0.20AOD for Terra, where AOD is the true AOD. However, a more accurate representation of
Terra's expected error is to account for the positive bias with asymmetrical error bounds:  -0.03 -
0.20AOD and +0.13 + 0.20AOD. The expected error bounds contain 2/3 of all AOD retrievals.
To assess the mean bias of the retrieval based on the retrieved AOD, we find that the mean bias
can  be  modeled  as  0.19  +  0.17*ln(AOD_MODIS+0.25)  for  Terra  and  0.15  +
0.14*ln(AOD_MODIS+0.25) for Aqua. Note that mean bias itself is subject to uncertainty.
We find a wide range of accuracy in the 3 km$^2$ product locally and regionally, with spatially
contiguous stations sometimes exhibiting significantly different validation statistics.    The
distribution of validation sites is highly skewed towards the northern mid-latitudes with over
50% of all collocations in the database resulting from these areas.  Within the northern mid-
latitudes, eastern North America and Europe with boreal Eurasia show some of the best
agreement with AERONET; however, western North America shows some of the poorest
agreement.  Regions outside of the northern mid-latitudes are less well represented in the
database, but we find that north/central South America including the Amazon, equatorial and
southern Africa, and Australia show good agreement with AERONET.  Mexico and the





Caribbean, southern South America, SW Asia, East Asia, and the maritime continent of
Southeast Asia generally show poor agreement. No attempt was made to isolate urban regions
from rural ones, or to otherwise sort the data by surface type.
The difference between MODIS-retrieved 3 $km^2$ AOD and AERONET observed values are
mostly independent of true AOD. This is unexpected as error bounds are defined as a function of
the percentage of AOD ($\pm0.05 \pm 0.20*$AOD).   However, the mean differences between MODIS
3 $km^2$ AOD and AERONET are dependent on AOD variability and availability with the more
variable the AOD, and the greater need for masking clouds and unfavorable surfaces in the
original retrieval, the higher the positive offset between MODIS and AERONET. Some of this is
due to the conditions of the original MODIS retrieval, and some is due to the difficulties of a
spatio-temporal match-up in the collocation methodology.   Interesting and unexplained is the
tendency for the differences between MODIS and AERONET to go negative when conditions
appear to be homogeneous and cloud-free.   We also find error dependencies on geometry, with
greater error in the far backscattering region and for Terra only, greater error in near-nadir views.
Some of these geometrical errors are introduced by uncertainties in the assumptions of surface
characteristics and aerosol optical properties in the MODIS retrieval, but the difference between
Terra and Aqua suggests differences in the sensors themselves.
We continue to see differences between the sensors in how validation statistics have evolved
over time. By limiting our time series analysis to only 26 AERONET stations that span the entire
time series, we eliminate changes in validation statistics due to changing AERONET station
distribution. We find that both sensors exhibit time series with strong seasonal dependence and
higher positive biases against AERONET in the northern spring and summer, than in northern
fall and winter, with Terra's positive bias always greater than Aqua's.  However, during the early
years of the time series, both sensors were reporting similar number of retrievals falling within
expected error. This changed during 2007-2009, when Terra's accuracy began to fall off and its
positive biases increased.  Aqua's bias against AERONET also increased during this time frame,
but not as rapidly as Terra's.   While, these drifts in validation accuracy suggest changes in
characterization accuracy of the MODIS sensors themselves, there are other factors.  The number
of collocations has fallen off towards the end of the time series.  We attribute this to a lag for





AERONET observations to be elevated to Level 2 status. Because of this lag, there may in fact be a change in the distribution of AERONET stations in the temporal collocation database, despite our best intentions, and this may introduce a temporal trend in the validation statistics. Furthermore, the aerosol system itself has undergone significant changes since 2000, with the U.S. and Europe drastically reducing their urban/industrial emissions and substituting wildfire smoke as their primary source of aerosol. Likewise emissions and resulting AOD from other regions experience both long-term trends and interannual variability. The combination of variations in AERONET station distribution and the changing aerosol system over the time series examined may be contributing to the trends seen in the validation statistics. However, the differences between Terra and Aqua are difficult to explain without pointing to sensor characterization stability.

The standard 10 km$^2$ product that meets expected error at 67% and 74% levels for Terra and Aqua, respectively, on a global basis is measurably more accurate than the 3 km$^2$ product examined here in detail. Similarly the global standard 10 km$^2$ AOD product exhibits half of the mean bias with Terra and no bias at all for Aqua. These validation statistics for the 10 km$^2$ standard product are preliminary and likely to change once a more comprehensive evaluation of the Collection 6, 10 km$^2$ product is completed. The 10 km$^2$ product numbers are provided here only to lend perspective to our results with the finer resolution product. Given this perspective, we confirm the Remer et al. (2013) recommendation that users whose interests are global should use the more robust and accurate 10 km$^2$ product, and leave the 3 km$^2$ product for specific applications that require the finer resolution representation of the AOD field.

This validation study only addressed the 3 km$^2$ AOD product over land, and did not evaluate the over water product. The study took a global and regional view, not a local one. Users of the product on a local level are encouraged to consider particular biases that may occur due to local conditions. For example, we know that the MODIS Collection 6 retrieval is systematically biased over urban surfaces. This is true for both the 10 km$^2$ and 3 km$^2$ products. This problem has been addressed and is substantially mitigated with the release of Collection 6.1 version of the algorithm (Gupta et al., 2016). In the meantime, our results here show that overall the MODIS Collection 6 algorithm is producing an AOD product at 3 km$^2$ resolution with sufficient accuracy





and with biases well-characterized so that it can be used quantitatively in a wide variety of
science and practical applications.
**7. Acknowledgement**
This work was supported by the NASA ROSES program NNH13ZDA001N-TERAQEA: Terra
and Aqua – Algorithms – Existing Data Products and NASA's EOS program managed by Hal
Maring. We thank MCST for their efforts to maintain and improve the radiometric quality of
MODIS data, and LAADS/MODAPS for the continued processing of the MODIS products. The
AERONET team (GSFC and site PIs) are thanked for the creation and continued stewardship of
the sun photometer data record; which is available from http://aeronet.gsfc.nasa.gov.

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



Table 1. Global statistics of comparison between MODIS 3 km AOD at 0.55 µm retrievals and
collocated AERONET observations for both Terra and Aqua, corresponding to three QA
categories (QAF=0 for poor quality, QAF=0,3 for all quality and QAF=3 for high quality). The
data used for the10km validation does not represent the same time period as 3km. MODIS-Terra
10km data period is 03/2000 to 06/2013 where as MODIS-Aqua 10km data period is 01/2003 –
6  06/2013.

| Sensor | MODIS-Terra | | | | MODIS-Aqua | | | |
|---|---|---|---|---|---|---|---|---|
| Resolution | 3 km | | | 10 km | 3 km | | | 10 km |
| QAF | 0 | 0, 3 | 3 | 3 | 0 | 0, 3 | 3 | 3 |
| N | 18055 | 112210 | 90162 | 82997 | 13935 | 89804 | 71248 | 66945 |
| R | 0.82 | 0.86 | 0.87 | 0.91 | 0.82 | 0.86 | 0.87 | 0.90 |
| Bias | 0.052 | 0.061 | 0.059 | 0.03 | 0.021 | 0.031 | 0.027 | 0.00 |
| Slope | 1.05 | 1.05 | 1.06 | 1.03 | 0.99 | 1.04 | 1.05 | 1.02 |
| RMSE | 0.18 | 0.15 | 0.15 | 0.11 | 0.16 | 0.14 | 0.13 | 0.10 |
| Within EE% | 52.56 | 59.62 | 62.47 | 68.68 | 58.55 | 66.08 | 68.36 | 74.38 |
| Above EE% | 35.03 | 33.50 | 31.33 | 24.27 | 25.17 | 23.51 | 21.47 | 15.42 |
| Below EE% | 12.42 | 6.88 | 6.2 | 7.04 | 16.28 | 10.42 | 10.18 | 10.21 |






Table 2. Regional statistics of inter-comparison between MODIS and AERONET. This is using
join data sets of Terra and Aqua for QAF=3 only.

| Region | N | Mean AOD | R | Bias | Slope | RMSE | Within EE% | Above EE% | Below EE% |
|---|---|---|---|---|---|---|---|---|---|
| N. American Boreal | 4136 | 0.111 | 0.93 | 0.079 | 1.39 | 0.14 | 56.14 | 43.76 | 0.10 |
| E. CONUS | 22450 | 0.129 | 0.90 | 0.029 | 1.22 | 0.09 | 73.41 | 19.74 | 6.85 |
| W. CONUS | 17645 | 0.096 | 0.68 | 0.116 | 1.45 | 0.19 | 45.34 | 53.27 | 1.39 |
| Central America | 2499 | 0.203 | 0.87 | 0.084 | 1.25 | 0.18 | 51.62 | 40.42 | 7.96 |
| South America | 5577 | 0.276 | 0.96 | -0.007 | 1.20 | 0.16 | 64.16 | 9.16 | 26.68 |
| S. South America | 5393 | 0.107 | 0.63 | 0.048 | 1.13 | 0.18 | 48.54 | 28.48 | 22.97 |
| Africa south of equator | 5849 | 0.184 | 0.81 | -0.020 | 0.71 | 0.10 | 68.44 | 12.84 | 18.72 |
| Equatorial Africa | 270 | 0.203 | 0.90 | 0.002 | 1.03 | 0.08 | 77.78 | 11.85 | 10.37 |
| Africa north of equator | 9870 | 0.302 | 0.83 | -0.039 | 0.63 | 0.18 | 61.00 | 18.50 | 20.50 |
| SW Asia | 405 | 0.220 | 0.78 | 0.164 | 1.25 | 0.21 | 33.58 | 66.17 | 0.25 |
| Europe - Mediterranean | 39792 | 0.162 | 0.79 | 0.043 | 1.06 | 0.11 | 70.62 | 24.63 | 4.75 |
| Eurasian Boreal | 13473 | 0.181 | 0.91 | 0.043 | 1.14 | 0.09 | 73.11 | 24.20 | 2.69 |
| East Asia Mid-Latitudes | 10009 | 0.370 | 0.91 | 0.110 | 1.09 | 0.20 | 56.03 | 41.22 | 2.75 |
| Peninsular Southeast Asia | 5259 | 0.501 | 0.91 | 0.039 | 1.05 | 0.18 | 68.09 | 22.02 | 9.89 |
| Indian Subcontinent | 8449 | 0.479 | 0.86 | 0.070 | 1.05 | 0.19 | 68.35 | 26.78 | 4.86 |
| Insular Southeast Asia | 853 | 0.243 | 0.85 | 0.118 | 1.03 | 0.20 | 50.41 | 48.30 | 1.29 |
| Australian Continent | 5965 | 0.087 | 0.59 | -0.021 | 0.57 | 0.08 | 69.52 | 8.92 | 21.56 |




Table 3. List of selected AERONET stations for the long-term analysis as presented in figure
2   9.

| Site Name | Latitude | Longitude |
|---|---|---|
| Canberra | -35.271 | 149.111 |
| Skukuza | -24.992 | 31.587 |
| Lake_Argyle | -16.108 | 128.749 |
| CUIABA-MIRANDA | -15.729 | -56.021 |
| Mongu | -15.254 | 23.151 |
| Jabiru | -12.661 | 132.893 |
| Chiang_Mai_Met_Sta | 18.771 | 98.972 |
| Kanpur | 26.513 | 80.232 |
| Izana | 28.309 | -16.499 |
| Saada | 31.626 | -8.156 |
| Nes_Ziona | 31.922 | 34.789 |
| TABLE_MOUNTAIN_CA | 34.380 | -117.680 |
| FORTH_CRETE | 35.333 | 25.282 |
| Blida | 36.508 | 2.881 |
| Cart_Site | 36.607 | -97.486 |
| Fresno | 36.782 | -119.773 |
| Evora | 38.568 | -7.912 |
| GSFC | 38.992 | -76.840 |
| KONZA_EDC | 39.102 | -96.610 |
| XiangHe | 39.754 | 116.962 |
| BSRN_BAO_Boulder | 40.045 | -105.006 |
| Lecce_University | 40.335 | 18.111 |
| Rome_Tor_Vergata | 41.840 | 12.647 |
| OHP_OBSERVATOIRE | 43.935 | 5.710 |
| Carpentras | 44.083 | 5.058 |
| Modena | 44.632 | 10.945 |






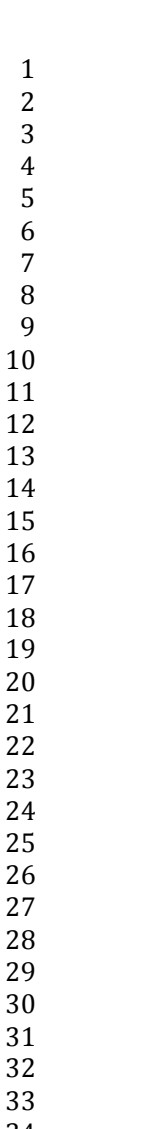

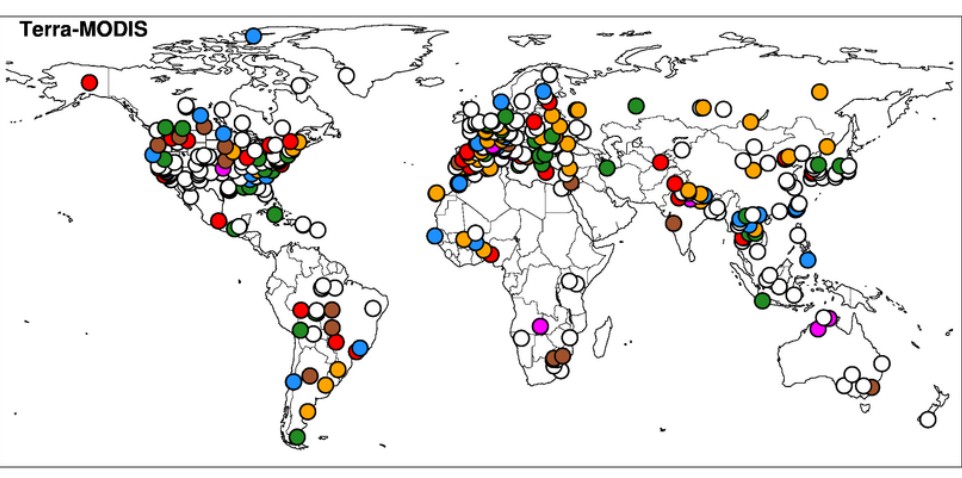

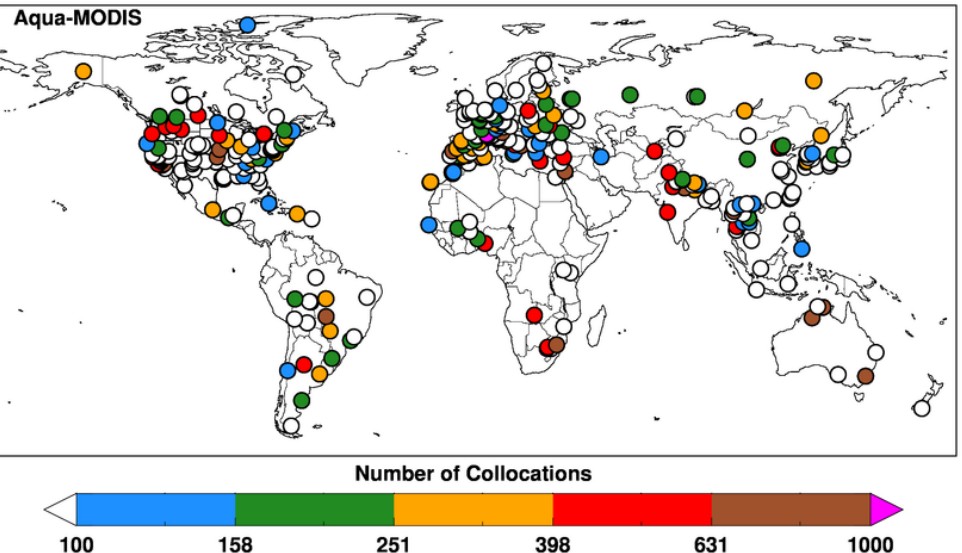

Figure 1. Locations of AERONET stations used in the validation study. The color scale shows the number of coincident MODIS-AERONET data points over each station for the entire period. Top panel is for MODIS-Terra and bottom panel for MODIS-Aqua. Most stations operated for only a subset of the 13 to 15 year record.

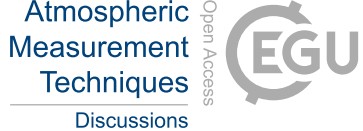



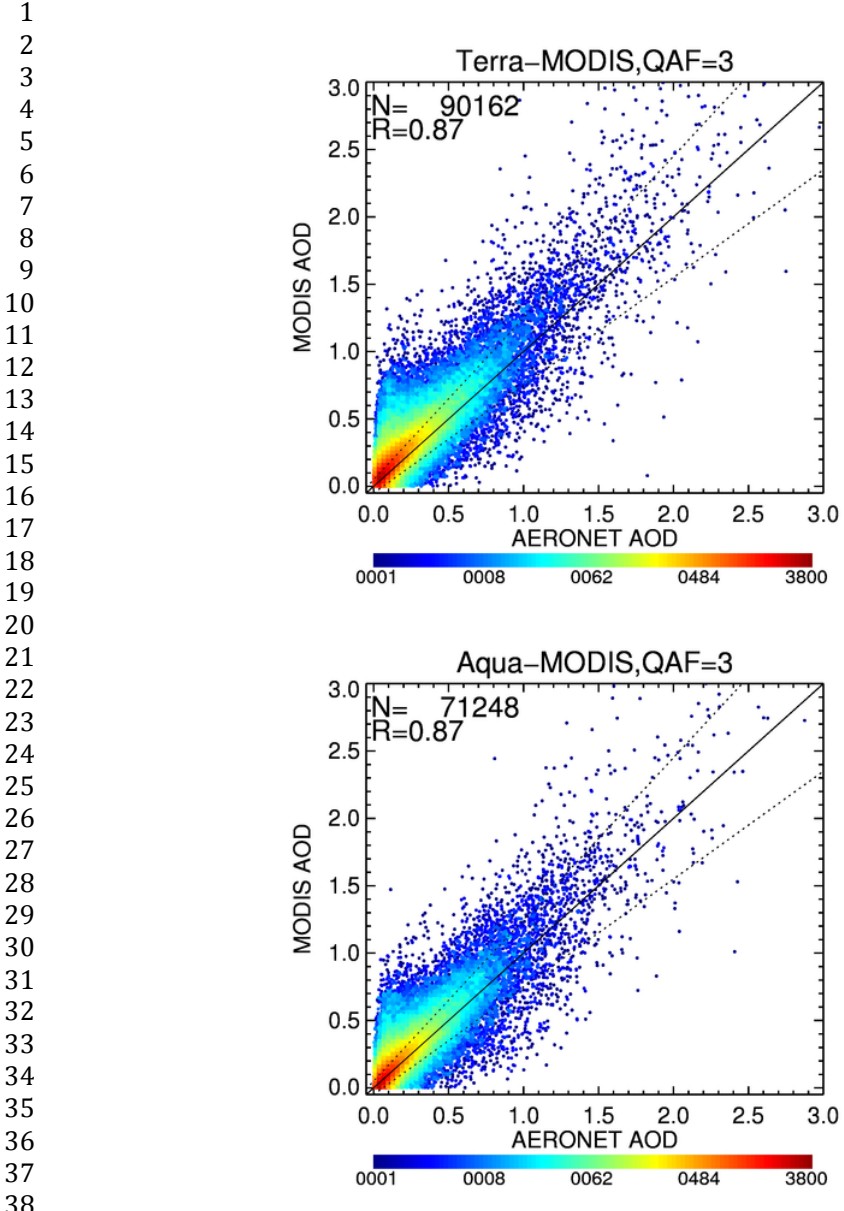

Figure 2. Two-dimensional density scatter plot of MODIS 3 km AOD versus AERONET
observed AOD at 0.55 µm for the global collocation data set. The top panel is for MODIS-
Terra for only the retrievals identified as 'high quality' (QAF=3), and the bottom panel is for
MODIS-Aqua for QAF=3. The solid line denotes the 1:1 line, and the dashed lines denote the
envelope of the expected error (EE), defined by Eq. 2.






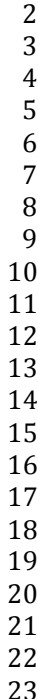
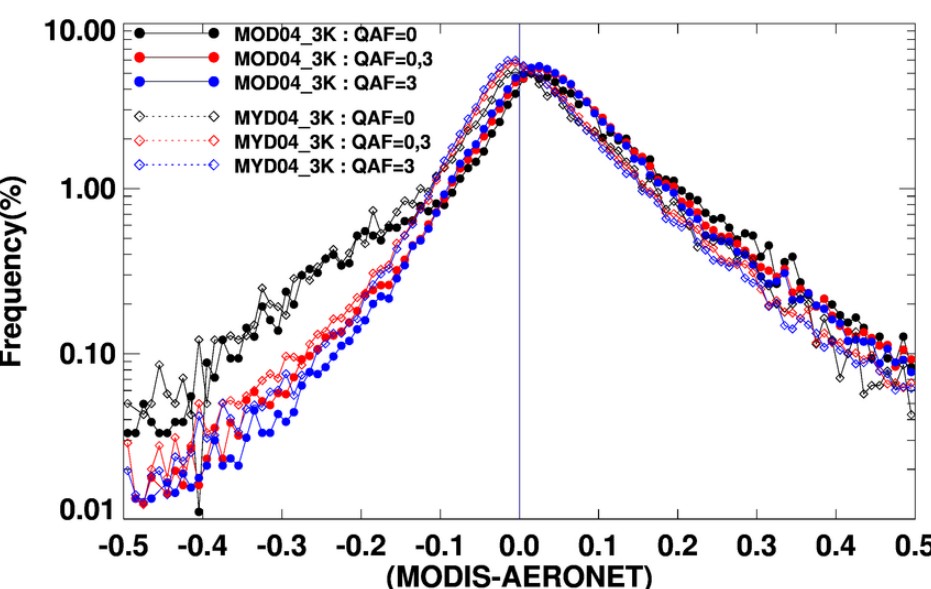

Figure 3. Global distribution of mean bias in MODIS 3 km$^2$ AOD retrievals with respect to
collocated AERONET observations. The circular dots with solid lines are for Terra values and
diamond dots with dotted lines are for Aqua values. The colors vary for the three quality levels
(QAF=0, poor quality; QAF=3, high quality; and QAF=0&3, all quality).

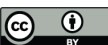





Figure 4. Statistics calculated from the collocation database at each AERONET station, individually, for Terra on the left and Aqua on the right. Shown are values for correlation coefficient (R), mean bias, percentage within expected error (EE%), and RMSE. Only stations with at least 100 collocations are plotted, which may differ between the two satellites, and only collocations with MODIS retrievals of QAF=3 are included.





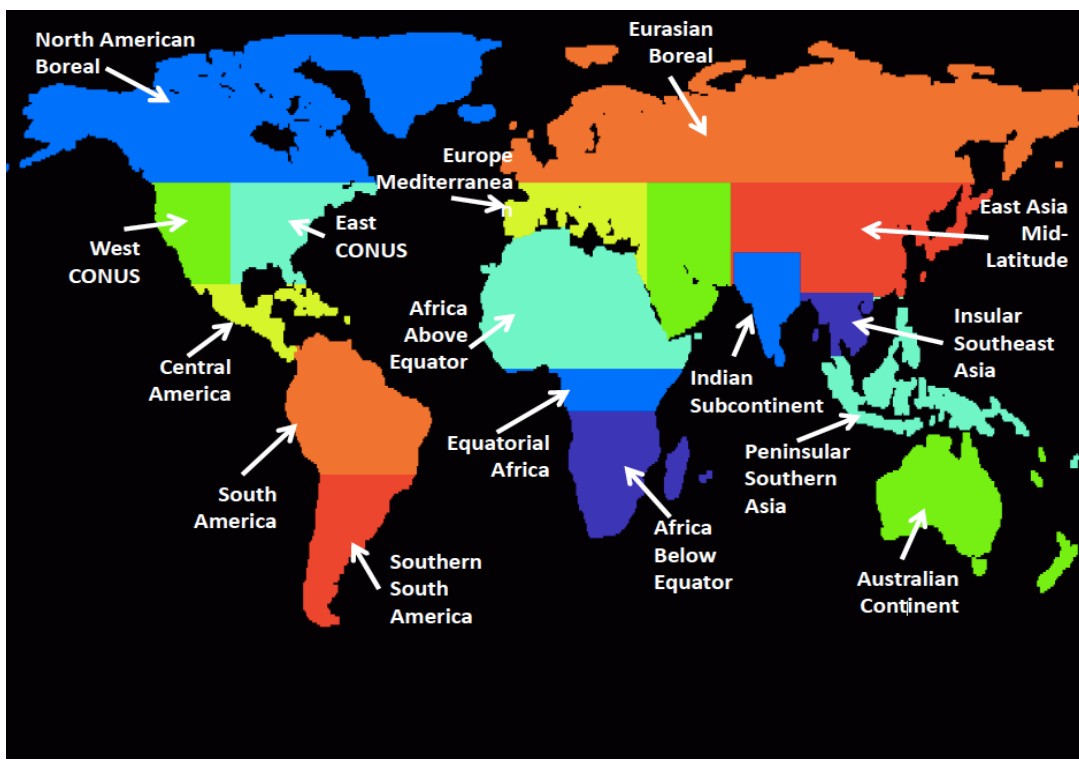

Figure 5. Map showing 17 selected parts of the world where regional analysis is performed.





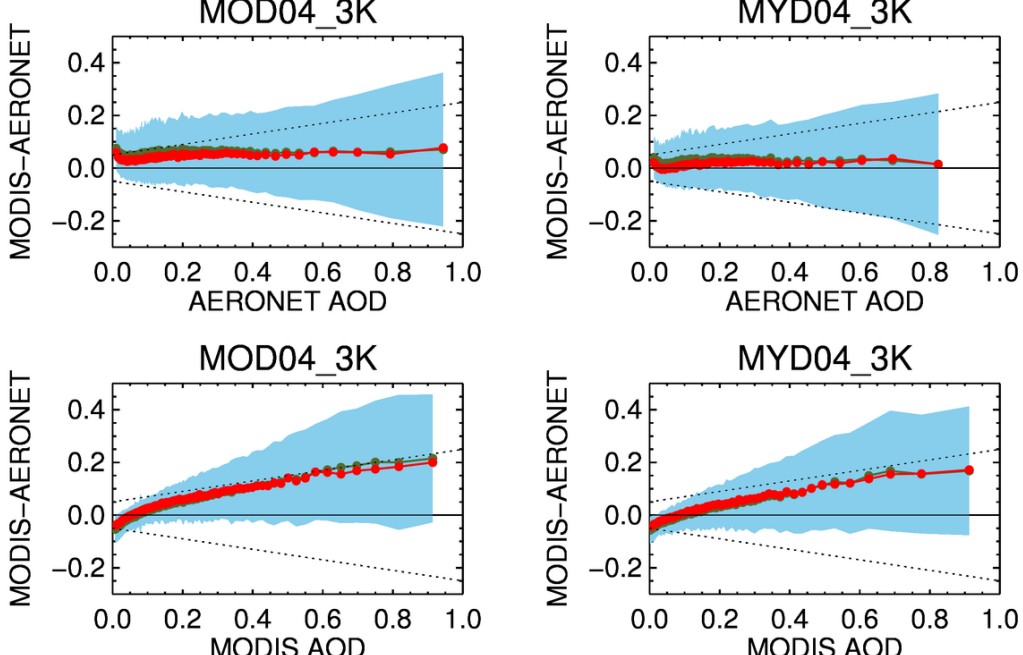

Figure 6. AOD differences between the MODIS 3 km product and AERONET for the global
collocation data set, QAF=3, as a function of AERONET AOD (top), MODIS AOD (bottom).
The left column shows Terra values and the right column shows Aqua.  The global data set was
sorted according to AOD, binned into bins with equal number of collocations, and then mean,
median and standard deviation of each bin was calculated.  Red dots and line show the mean.
Black dots and line show the median.  The blue cloud indicates one standard deviation of each
bin. The horizontal black line denotes zero difference, and the dashed lines indicate EE
envelopes.  Positive values indicate that MODIS AOD is higher than AERONET.





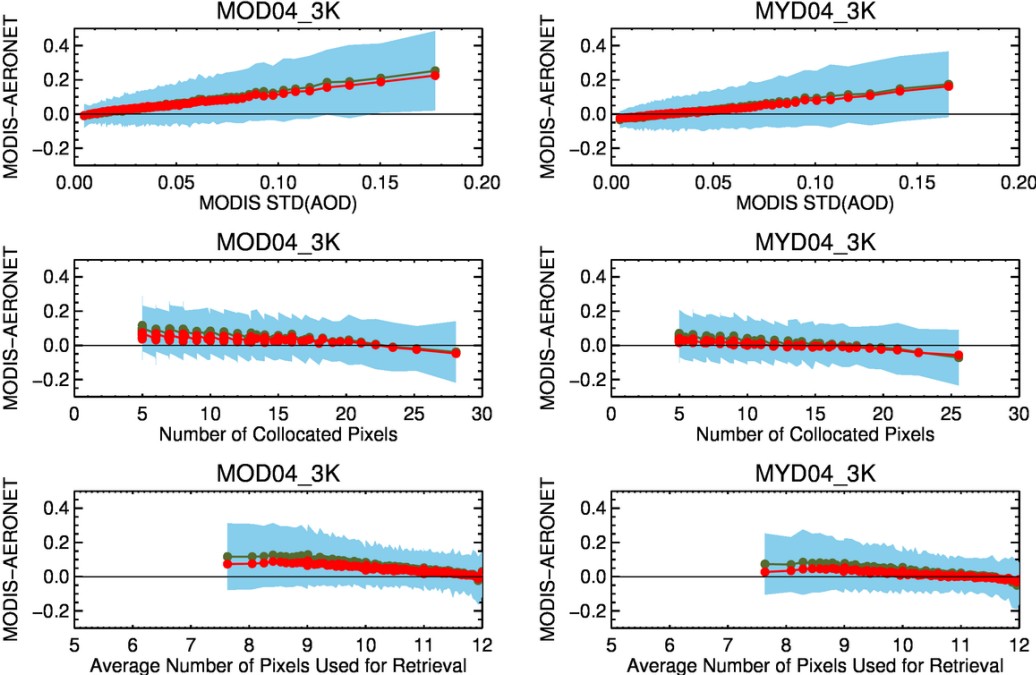

Figure 7.   Same as Figure 6 except for standard deviation of MODIS AOD within the 5x5
collocation box (top row), number of MODIS retrievals with the 5x5 collocation box (middle
row), and number of MODIS reflectance native pixels used by the retrieval, averaged for all
retrievals made in the 5x5 collocation box (bottom row).



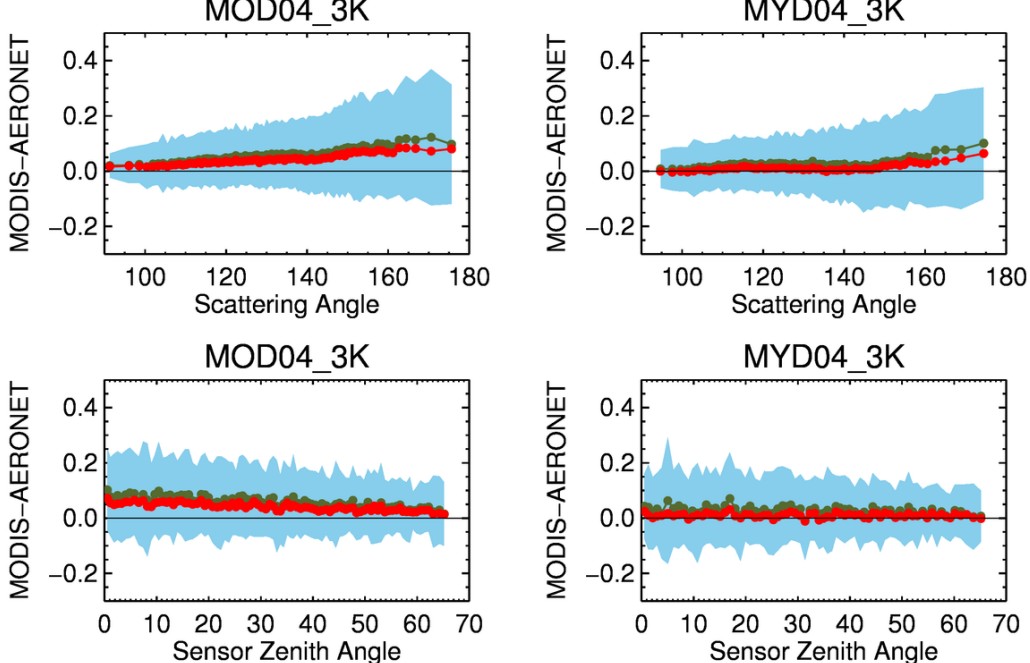

Figure 8.  Same as Figure 6 except for scattering angle (top) and sensor zenith angle (bottom).

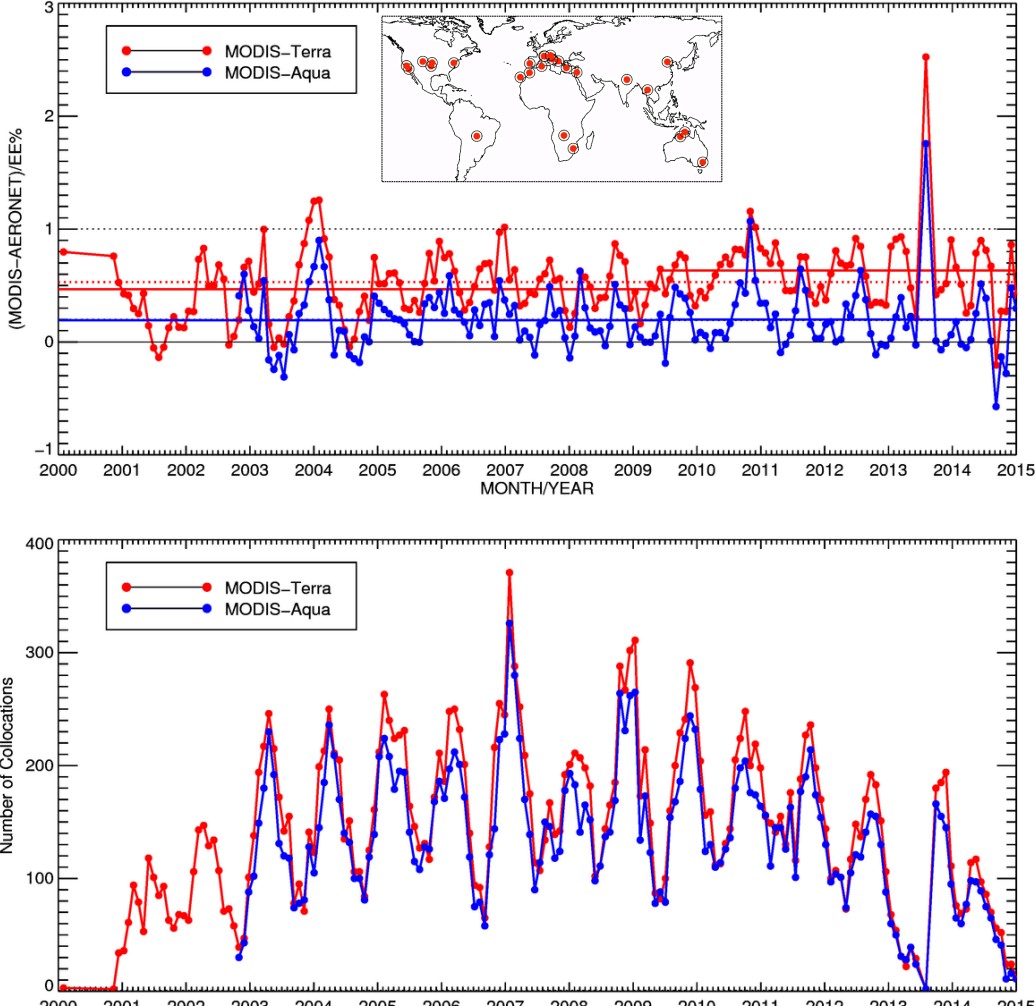

Figure 9.  Time series of monthly mean error ratios (Eq. 3) (top), and number of collocations
(bottom) for the global collocation data set from 26 selected long-term AERONET stations.  The
Terra record is in red, and Aqua in blue.  Note Aqua's record begins two years after Terra, and
the total number of collocations is temporally variant.   Only MODIS 3 km retrievals with
QAF=3 are included. The horizontal red and blue lines are temporal means of ER.  Dotted red
and blue horizontal lines indicate long-term temporal mean ER for each satellite.  Solid red lines
are temporal mean ER calculated for Terra for two periods (2000-2010) and  (2011-2015). The
map in the inset shows locations of AERONET stations used in this analysis with more details
provided in table 3