# Peer review of "Validation of MODIS 3 km Land Aerosol Optical Depth from NASA's EOS Terra and Aqua Missions"

_Atmospheric Measurement Techniques, 2018_

## Referee Comment (RC1) · Anonymous Referee #1 · 13 Mar 2018

Review of Gupta et al., "Validation of MODIS 3 km Land Aerosol Optical Depth from NASA's EOS Terra and Aqua Missions", submitted for publication in AMT

General Comments

MODIS Terra and Aqua Aerosol Optical Depth (AOD) retrievals from Dark Target (DT) land algorithm were globally validated. For this, AEORNET V2 L2.0 and MODIS AOD at 550 nm were used. MODIS AOD retrievals were averaged for 5 x 5 spatial window centered at the AERONET station and AERONET measurements were averaged for ± 30 minutes of satellite overpass time. Total 90,162 and 71,248 high-quality collections were collected for Terra and Aqua, respectively. The quality of collocations was evaluated using correlation coefficient, regression slope, Mean Bias (MB), Root Mean Square Error (RMSE), Expected Error (EE) is defined by Remer et al. (2013), and

Error Ratio (ER). Overall, MODIS Terra and Aqua AOD retrievals are highly correlated with AERONET AOD, and 62.5% and 68.4% of AOD retrieved fall within the EE, respectively. The manuscript is well written and has a merit for publication in AMT, but some proofreading is required for small technical errors.

Specific Comments

L14-20: These lines are more suitable in the introduction section than here.

L20: It is recommended to avoid the use of the first pronoun is scientific writings.

L28-29: Please mention Ångström exponent value ($\alpha$440-675?).

L30: AOD is interpolated to 0.55 $\mu$m > AOD is interpolated to 0.55 $\mu$m using Ångström exponent ($\alpha$440-675?).

Technical Corrections

L16: dark target > Dark Target (DT)

L17: aerosol optical depth > Aerosol Optical Depth (AOD)

L22: AERONET > AErosol RObotic NETwork (AERONET)

L23: MODIS Terra > MODIS-Terra

L24: 62..5% > 62.5%

L26: (0.05+0.2*AOD) > (0.05 + 0.2 $\times$ AOD)

L27: RObotoic > RObotic

L7, 22: 10 km2 > 10 km

L8: 0.5 km2 > 0.5 km

L9, 14, 18, 22: 3 km2 > 3 km, please correct everywhere in the manuscript.

L15: 3km > 3 km

Page 7:

L1: level 2.0, version 2.0 > Version 2 Level 2.0

L8: 3km > 3 km

L14: 50x50 km2 > 50 $\times$ 50 km2

L14: x > $\times$, please correct everywhere in the manuscript.

L17, 20, 26: $\pm$30 > $\pm$ 30

L20: $\pm$ 30 minutes of overpass > $\pm$ 30 minutes satellite overpass

L5: AEROENT values > AERONET values (Figure 2)

L7: Delete "Results are plotted in Figure 2".

L16: QAF=0 > QAF=0 (Table 1?)

L30: R$\geq$0.78 > R $\geq$ 0.78

L8: 75% > 70%?

L8: Delete "there"

L1: AOD (<0.1) > AOD (< 0.10)

L6: biases of >0.10 > biases of > 0.10

L16, 27: 5x5 > 5 × 5

L23: MODIS – AERONET > MODIS-AERONET

L10: $-1 \leq ER \leq 1$ > $-1 \leq ER \leq 1$

L29: Only Level 2, quality assured > Version 2 Level 2.0, cloud screened and quality assured

---

## Short Comment (SC1) · 14 Mar 2018

A. M. Sayer

andrew.sayer@nasa.gov

This Short Comment concerns the linear least-squares regression results presented in the manuscript (in e.g. Table 1). I am posting it here after discussion with the authors in person.

While it is a commonly-used technique, unfortunately AOD data of this type are generally not suitable for the use of ordinary least squares linear regression. The technique requires certain assumptions about the nature of the data to be able to provide quantitatively meaningful regression characteristics (and uncertainties on those characteristics), and these assumptions are all questionable or violated in the case of remotely-sensed AOD data of this type. For example, assumptions of linearity, independence

of data points, existence of a single population, Gaussian behaviour of residuals, and scale-independence of AOD uncertainties. The result is that the output numbers are not meaningful in the sense that we want to use them. It is not a matter of the results being noisy; they can be systematically biased or in some cases meaningless.

I acknowledge that it is a commonly-used technique but that should not in my view be a valid justification for doing something which is statistically inappropriate in a scientific journal. It is best for us to stop doing it and in this way hopefully spread good practice more broadly through the community.

The reason least-squares linear regression is a popular choice is it gives us two parameters (intercept and slope) with which we can say something about what biases/offsets are in the limiting cases of low-AOD and high-AOD regimes. The question then is what is the best way to convey this type of information in a more statistically-appropriate way?

Fortunately the authors have largely already done so. Since we typically frame our retrieval performance in terms of fraction within expected error (EE), the authors' inclusion of summaries of what proportion of matchups are below, within, and above the EE is one welcome step. Another is with the binned type of plots seen within e.g. Figure 6 (which incidentally already shows that the relationships are overall not linear). The values of the offset for the low-AOD bins provide an indication of typical biases in low-AOD conditions. And the relative magnitudes of the offset for the high-AOD bins provide an indication of typical biases in high-AOD conditions. Or if there is no apparent AOD-dependence then you can just state that the offset appears invariant with AOD. I suggest that the authors remove least-squares slope and intercepts results from the paper. For the same reason, ideally Pearson's linear correlation coefficient could also be replaced with Spearman's rank correlation coefficient. If the authors wish to include replacement information instead of slope/intercept to summarise the global statistics, I suggest adding something like the magnitude and sign of absolute bias as seen in the low-AOD bins, and the relative magnitude of the bias from the high-AOD bins.

[Figure]

For example, eyeballing from the bottom-left panel of Figure 6 (Terra, defined relative to MODIS AOD), when MODIS retrieves AOD in the range -0.05 to 0 it looks like the typical offset is about -0.05. When MODIS retrieves AOD above about 0.4, it looks like the bin mean/median bias are positive and about 20%. So in this case you might say that the typical biases are around -0.05 in the cleanest conditions and +20% in high-AOD conditions. Or if you take the top-left panel (Terra, defined relative to AERONET AOD), it looks like the bias it looks like the typical bias is around 0.05-0.1 regardless of AOD. In my view those numbers are more appropriate and more useful statistics to report than the regression slope/intercept.

---

## Referee Comment (RC2) · Anonymous Referee #2 · 24 Mar 2018

General Comments:

The manuscript summarizes the results from a global, long-term (12-15 yr) evaluation of the MODIS Collection 6 dark target 3km aerosol optical depth product, using Level 2 AOD from AERONET sites. A large number of MODIS-AERONET collocations (161,410) with highest quality flag (QA=3) from many regions are used to study overall MODIS 3km AOD performance on global and regional scales, time series of AOD performance, and factors influencing AOD performance. The result is a follow-up to previous studies of MODIS C6 10km AOD product (Levy et al., 2013) and 3km product (Remer, 2013), with this product providing a more thorough global evaluation of C6 3km product performance and some comparisons with C6 10km AOD product. The paper provides a thorough and well-documented source of information regarding MODIS

C6 3km DT performance, including caveats for its usage. Methods are clearly explained and the analysis is thorough and pedagogically-sound. Scientific significance and scientific quality are very good and the paper meets the standards for publication in AMT but the authors should first clean up the document for persistent grammatical and sentence structure errors, which impact readability in many places. There is also redundancy in many places (a few of which I list below and recommend changes for) but I find some of the redundancies beneficial.

Specific Comments:

1. Section 2. Page 5. Lines 9-10. The authors state that "Therefore, for the 3 km2 product, any fewer than 5 native 10 pixels automatically receives QAF=0. QAF values assigned as 1 or 2 are based on other criteria.". Please either mention these criteria or reference a paper where the user can obtain such information. 2. Page 10. Lines 19-21. Provide some reasoning as to why correlation breaks down at these sites. Grammar could also be improved upon in this sentence. You may wish to state that "Correlation is weaker" or something along these lines, instead of "Correlation breaks down..". 3. P. 19. Lines 4-6. The authors state that "Furthermore, the aerosol system itself has undergone significant changes since 2000, with the U.S. and Europe drastically reducing their urban/industrial emissions and substituting wildfire smoke as their primary source of aerosol." This is likely true for the western U.S. but not likely to be true for the eastern U.S. Authors should either specify 'western U.S.' or provide the results from some studies (which I have not seen) supporting their assertion. Regardless-they should cite some studies which substantiate this claim. 4. P.19, Lines 27-28: Please cite reference(s) to support this claim so that the interested reader can view these paper(s). Also state whether this systematic bias in MODIS C6 holds true for both DT and DB (as is implied by not stating which), DT only, or DB only.

Technical Corrections:

There are many grammatical errors and incorrect sentence structure exists throughout

the document. Readability and flow of the manuscript will be greatly improved upon once these are fixed. For brevity, I only list a few but encourage the authors to review the grammar and fix accordingly, or else ask an outsider to review the manuscript for grammar, sentence structure, and readability. There are examples of incorrect sentence structure (missing commas, commas placed where new sentences should begin, ….). More efficient wording should also be utilized in many places, in place of long, rambling sentences.

P. 6 Lines 14-15. Grammar. It should read as "AERONET processes these spectral measurements to derive AOD at the wavelengths corresponding to the direct sun measurements."

P. 7 Line 6: Please add the phrase "AOD at 550 nm" to the phrase "We have created a collocated data set (CDS) of both MODIS-Terra and MODIS-Aqua" to qualify the measurements being compared. It is obvious to most readers but still should be explicitly stated.

P. 9. Lines 14-17: "Identified retrieval quality matters to product accuracy with QAF=3 showing stronger correlation, smaller RMSE and more retrievals falling within expected error than QAF=0, but the high quality data set loses about 20% of the retrievals." Please fix grammar. One suggestion is to replace "matters to" with "influences".

P.10. Line 31. Change "can report" to "often report" or similar.

P. 10 Lines 6-10. The sentence is too long and difficult to follow. It should be broken down into two 2-3 sentences. An alternative is to enumerate the stated factors influencing regionally-specific retrieval performance. This alone would improve readability. Also change the word "will" in line 6 to "is". There are several other places in the manuscript with similar long, rambling sentences that would be easier to follow if broken down into shorter, clear sentences.

P.11. Line 12: Change the word "fades" to "ranges from" .

P.11 Lines 13-14. Please reword the sentence "For many of the stations, positive mean biases decrease from Terra to Aqua." to something along the lines of "At many of the stations, the positive mean AOD bias is larger for Terra than for Aqua.".

P.11. Line 17. You mix present and future tense throughout the paper. Please pick a tense and stick with it. Present tense is typically used when describing the current study (yours) and past tense is typically used to describe the referenced work of others. For this reason, I recommend using present tense throughout the paper.

P.11. Line 24. Delete the sentence "Only QAF=3 retrievals are included.". This has already been mentioned.

P.11 Lines 17-24. Please combine the two short paragraphs with 2 sentences each into a single paragraph.

P. 11 Line 26 through P.12 Line 17. These two paragraphs contain numerous redundancies and could easily be combined into a single paragraph. One example is on P.12. Lines 5-7. This sentence has already been stated above and should be eliminated. You could also include the good agreement for the sites in "north/central South America, equatorial and southern Africa, and Australia" in the last sentence of previous paragraph but re-stating that "Regions where MODIS 3 km2 retrievals exhibit especially good agreement with AERONET 6 collocations include E. CONUS and Europe" is unnecessary. There are also redundant statements made throughout the paragraph, which could easily be consolidated with the previous paragraph.

P.12 Line 19. Please either change the 3 km^2 to 3km or specify it as 3 x 3 km2 throughout the document. You do so in the abstract but not in the other sections.

P.12. Lines 22-27. Please fix several grammatical errors.

P.14 Lines 9-13. This repeats what was already stated in

P. 15. Lines 25-27: Another case of 2-sentence paragraph. Please combine with one of adjacent paragraphs

P.17 Lines 11-12. The authors state that "There is significant degradation of validation accuracy if MODIS retrievals of Poor data quality (QA<3) are included in the analysis". This implies that QA values of 1 and 2 were used and that they gave "poor quality". I thought that only QA=3 and QA=0,3 were used. If I am correct, please change QA<3 to QA=0 to accurately describe the data used. Also please be consistent in the acronym for quality flag. You use QA in some places and QAF in others. Please pick one of them and use throughout the document.

P. 17 Line 14. Please correct the dimensions of MODIS product. You state it as '3km2'. Please fix here and other instances in the paper.

P 17 Line 23 – P. 18 Line 3: There are several grammatical errors in this paragraph (and similar errors in other sections of document), including missing commas and similar errors. Please fix throughout the document.

P. 19. Lines 13-15. This is one of many sentences throughout the document which needs the grammar fixed. There are commas places where they should not be placed and missing from places where they should appear.
* * *

---

## Author Response (AR1)

**Anonymous Referee #1 (doi:10.5194/amt-2018-44-RC1, 2018)**
**General Comments**
MODIS Terra and Aqua Aerosol Optical Depth (AOD) retrievals from Dark Target (DT) land
algorithm were globally validated. For this, AEORNET V2 L2.0 and MODIS AOD at 550 nm
were used. MODIS AOD retrievals were averaged for 5 x 5 spatial window centered at the
AERONET statio and AERONET measurements were averaged for ± 30 minutes of satellite
overpass time. Total 90,162 and 71,248 high-quality collections were collected for Terra and
Aqua, respectively. The quality of collocations was evaluated using correlation coefficient,
regression slope, Mean Bias (MB), Root Mean Square Error (RMSE), Expected Error (EE) is
defined by Remer et al. (2013), and Error Ratio (ER). Overall, MODIS Terra and Aqua AOD
retrievals are highly correlated with AERONET AOD, and 62.5% and 68.4% of AOD retrieved
fall within the EE, respectively. The manuscript is well written and has a merit for publication in
AMT, but some proofreading is required for small technical errors.
**We thank the reviewer for the review and presenting his/her stylistic suggestions.  We have**
**considered each one carefully. All our responses are in BOLD.**
Specific Comments
Page 1 L14-20: These lines are more suitable in the introduction section than here.
**We have revised the abstract and removed these lines.**
L20: It is recommended to avoid the use of the first pronoun is scientific writings.
**There are many writing guidelines that encourage the use of active voice in a scientific**
**publication, which at times requires the use of a first person pronoun.  These guidelines**
**include the journal Nature:**
**Nature journals prefer authors to write in the active voice ("we performed the**
**experiment...")**
**https://www.nature.com/authors/author_resources/how_write.html**
**the journal Science:**
**Use active voice when suitable, particularly when necessary for correct syntax (e.g., "To**
**address this possibility, we constructed a λZap library . . .," not "To address this**
**possibility, a λZap library was constructed . . .").**
**http://www.sciencemag.org/site/feature/contribinfo/prep/res/style.xhtml**
**and others.  For *Atmospheric Measurements and Techniques* there are no similar guidelines**
**towards active versus passive voice.  Under English guidelines and house standards, AMT**

**only states, "We accept all standard varieties of English in order to retain the author's voice."  That statement is made mostly to address variants in spellings of specific words.  However, there is obviously enough leeway in the journal's style guideline and enough variation across scientific fields to accommodate the active voice and first person pronouns in AMT publications.  As readers we much prefer the crisp style of active voice in scientific writing, and as writers, we respectfully choose to continue to write in this style.**

Page 6 L28-29: Please mention Ångström exponent value (α440-675?).

**The reviewer may be mistaken.  We have not used Angstrom exponent in our interpolation but used actual AOD values in multiple channels using method described by Eck et al., 1999.  Angstrom exponent assumes linearity of the AOD vs. wavelength relationship in log-log space.  Eck et al., 1999 showed that there is often curvature in the relationship, and therefore a more accurate interpolation between wavelengths takes into account this curvature.  There are a great many papers that cite Eck et al., 1999 (over 1000 on Google Scholar) that have bypassed Angstrom exponent and gone to the non-linear interpolation, and therefore we do not see a need to explain our reasons for making this bypass.**

Page 16 L30: AOD is interpolated to 0.55 μm > AOD is interpolated to 0.55 μm using Ångström exponent (α440-675?).

**See above comment.**

Technical Corrections
**Page 1:**
L16: dark target > Dark Target (DT)
**REVISED**
L17: aerosol optical depth > Aerosol Optical Depth (AOD)
**REVISED**
L22: AERONET > AErosol RObotic NETwork (AERONET)
**REVISED**
L23: MODIS Terra > MODIS-Terra
**REVISED**
L24: 62..5% > 62.5%
**REVISED**
L26: (0.05+0.2*AOD) > (0.05 + 0.2 × AOD)
**REVISED**
Page 3 L27: RObotoic > Robotic
**REVISED**
Page 4 L7, 22: 10 km2 > 10 km
**REVISED**
L8: 0.5 km2 > 0.5 km
**REVISED**
L9, 14, 18, 22: 3 km2 > 3 km, please correct everywhere in the manuscript.
**REVISED Everywhere**
L15: 3km > 3 km

**REVISED**
**Page 7:**
L1: level 2.0, version 2.0 > Version 2 Level 2.0
**REVISED**
L8: 3km > 3 km
**REVISED**
L14: 50x50 km2 > 50 × 50 km2
**REVISED**
L14: x > ×, please correct everywhere in the manuscript.
**REVISED**
L17, 20, 26: ±30 > ± 30
**REVISED**
L20: ± 30 minutes of overpass > ± 30 minutes satellite overpass
**REVISED**
**Page 9:**
L5: AEROENT values > AERONET values (Figure 2)
L7: Delete "Results are plotted in Figure 2".
**REVISED**
L16: QAF=0 > QAF=0 (Table 1?)
**REVISED**
**Page 11:**
L30: R≥0.78 > R ≥ 0.78
**REVISED**
**Page 12:**
L8: 75% > 70%?
**REVISED**
L8: Delete "there"
**REVISED**
**Page 13:**
L1: AOD (<0.1) > AOD (< 0.10)
**REVISED**
L6: biases of >0.10 > biases of > 0.10
**REVISED**
L16, 27: 5x5 > 5 × 5
**REVISED**
**Page 14:**
L23: MODIS – AERONET > MODIS-AERONET
**REVISED**
**Page 15:**

L10: -1≤ER≤1 > $-1 \le ER \le 1$
**REVISED**

**Page 16:**
L29: Only Level 2, quality assured > Version 2 Level 2.0, cloud screened and quality
Assured
**REVISED**

**Anonymous Referee #2 doi:10.5194/amt-2018-44-RC2, 2018**

**General Comments:**

The manuscript summarizes the results from a global, long-term (12-15 yr) evaluation of the MODIS Collection 6 dark target 3km aerosol optical depth product, using Level 2 AOD from AERONET sites. A large number of MODIS-AERONET collocations (161,410) with highest quality flag (QA=3) from many regions are used to study overall MODIS 3km AOD performance on global and regional scales, time series of AOD performance, and factors influencing AOD performance. The result is a follow-up to previous studies of MODIS C6 10km AOD product (Levy et al., 2013) and 3km product (Remer, 2013), with this product providing a more thorough global evaluation of C6 3km product performance and some comparisons with C6 10km AOD product. The paper provides a thorough and well-documented source of information regarding MODIS C6 3km DT performance, including caveats for its usage. Methods are clearly explained and the analysis is thorough and pedagogically-sound. Scientific significance and scientific quality are very good and the paper meets the standards for publication in AMT but the authors should first clean up the document for persistent grammatical and sentence structure errors, which impact readability in many places. There is also redundancy in many places (a few of which I list below and recommend changes for) but I find some of the redundancies beneficial.

**We thank the reviewer for the review. We have considered each one carefully. All our responses are in BOLD.**

**Specific Comments:**
1. Section 2. Page 5. Lines 9-10. The authors state that "Therefore, for the 3 km2 product, any fewer than 5 native 10 pixels automatically receives QAF=0. QAF values assigned as 1 or 2 are based on other criteria.". Please either mention these criteria or reference a paper where the user can obtain such information.

**Remer et al., 2013 Reference for further details is added. We have also revised the text with more explanation.**

2. Page 10. Lines 19-21. Provide some reasoning as to why correlation breaks down at these sites. Grammar could also be improved upon in this sentence. You may wish to state that "Correlation is weaker" or something along these lines, instead of "Correlation breaks down..".

**Revision has been made in the text. Reasoning for the weaker correlations is given.**

3. P. 19. Lines 4-6. The authors state that "Furthermore, the aerosol system itself has undergone significant changes since 2000, with the U.S. and Europe drastically reducing their urban/industrial emissions and substituting wildfire smoke as their primary source of aerosol." This is likely true for the western U.S. but not likely to be true for the eastern U.S. Authors should either specify 'western U.S.' or provide the results from some studies (which I have not seen) supporting their assertion. Regardless-they should cite some studies which substantiate this claim.

**The point is that there has been a drastic reduction in traditional urban/industrial aerosol**
**types throughout the U.S. and Europe.  When there is a void, other types of aerosol become**
**more important.  Even in eastern U.S. there have been many intrusions of transported**
**wildfire smoke from the west and from Canada, for example.  However, we do see the**
**reviewer's point here and have modified the statement and have added three references**

**Karnieli, A., Y. Derimian, R. Indoitu, N. Panov, R. C. Levy, L. A. Remer, W. Maenhaut,**
**and B. N. Holben (2009), Temporal trend in anthropogenic sulfur aerosol transport from**
**central and eastern Europe to Israel, J. Geophys. Res., 114, D00D19,**
**doi:10.1029/2009JD011870.**

**Toon, O.B., et al., Planning, implementation, and scientific goals of the Studies of Emissions**
**and Atmospheric Composition, Clouds and Climate Coupling by Regional Surveys**
**(SEAC4RS) field mission, J. Geophys. Res. Atmos., 121, doi:10.1002/2015JD024297, 2016.**

**Hand, J.L., B.A. Schichtel, W.C. Malm, S. Copeland, J.V. Molenar, N. Frank, M.**
**Pitchford, Widespread reductions in haze across the United States from the early 1990s**
**through 2011., Atmos. Environ., 94, 671-679, 2014.**
**https://doi.org/10.1016/j.atmosenv.2014.05.062**

4. P.19, Lines 27-28: Please cite reference(s) to support this claim so that the interested reader
can view these paper(s). Also state whether this systematic bias in MODIS C6 holds true for both
DT and DB (as is implied by not stating which), DT only, or DB only.

**Gupta et al., 2016 provides further details and proposed changes in the algorithm to**
**correct biases over urban surface, which are now implemented in the Collection 6.1 data**
**sets. Reference is added.**

**This is true for dark target only. The text has been revised to make this clear.**

**Technical Corrections:**
There are many grammatical errors and incorrect sentence structure exists throughout the
document. Readability and flow of the manuscript will be greatly improved upon once these are
fixed. For brevity, I only list a few but encourage the authors to review the grammar and fix
accordingly, or else ask an outsider to review the manuscript for grammar, sentence structure,
and readability. There are examples of incorrect sentence structure (missing commas, commas
placed where new sentences should begin, . . ..). More efficient wording should also be utilized
in many places, in place of long, rambling sentences.

P. 6 Lines 14-15. Grammar. It should read as "AERONET processes these spectral
measurements to derive AOD at the wavelengths corresponding to the direct sun measurements."

**REVISED**.

P. 7 Line 6: Please add the phrase "AOD at 550 nm" to the phrase "We have created a collocated data set (CDS) of both MODIS-Terra and MODIS-Aqua" to qualify the measurements being compared. It is obvious to most readers but still should be explicitly stated.

**REVISED**

P. 9. Lines 14-17: "Identified retrieval quality matters to product accuracy with QAF=3 showing stronger correlation, smaller RMSE and more retrievals falling within expected error than QAF=0, but the high quality data set loses about 20% of the retrievals." Please fix grammar. One suggestion is to replace "matters to" with "influences".

**REVISED**

P.10. Line 31. Change "can report" to "often report" or similar.

**Changed to "sometimes report"**

P. 10 Lines 6-10. The sentence is too long and difficult to follow. It should be broken down into two 2-3 sentences. An alternative is to enumerate the stated factors influencing regionally-specific retrieval performance. This alone would improve readability. Also change the word "will" in line 6 to "is". There are several other places in the manuscript with similar long, rambling sentences that would be easier to follow if broken down into shorter, clear sentences.

**The text has been revised for clarity and grammar.**

P.11. Line 12: Change the word "fades" to "ranges from".

**'fades' replaced with 'decreases'. We are comparing Terra and Aqua biases here. Words added to clarify this point.**

P.11 Lines 13-14. Please reword the sentence "For many of the stations, positive mean biases decrease from Terra to Aqua." to something along the lines of "At many of the stations, the positive mean AOD bias is larger for Terra than for Aqua."

**REVISED**

P.11. Line 17. You mix present and future tense throughout the paper. Please pick a tense and stick with it. Present tense is typically used when describing the current study (yours) and past tense is typically used to describe the referenced work of others. For this reason, I recommend using present tense throughout the paper.

**REVISED**

P.11. Line 24. Delete the sentence "Only QAF=3 retrievals are included.". This has already been mentioned.

**REVISED**

P.11 Lines 17-24. Please combine the two short paragraphs with 2 sentences each into a single paragraph.

**REVISED**

P. 11 Line 26 through P.12 Line 17. These two paragraphs contain numerous redundancies and could easily be combined into a single paragraph. One example is on P.12. Lines 5-7. This sentence has already been stated above and should be eliminated. You could also include the good agreement for the sites in "north/central South America, equatorial and southern Africa, and Australia" in the last sentence of previous paragraph but re-stating that "Regions where MODIS 3 km2 retrievals exhibit especially good agreement with AERONET 6 collocations include E. CONUS and Europe" is unnecessary. There are also redundant statements made throughout the paragraph, which could easily be consolidated with the previous paragraph.

**We have removed an entire paragraph because of the redundancy between the discussion concerning the analysis at the local station level and that of the regional level. We do want to keep the statements about CONUS and Europe because we want to point out that the global statistics are heavily weighted by the collocations in this limited part of the globe. This can be implied by the circles plotted in Figure 5, but are not as apparent as when the number of collocations are tabulated by region in Table 1.**

P.12 Line 19. Please either change the 3 km^2 to 3km or specify it as 3 x 3 km2 throughout the document. You do so in the abstract but not in the other sections.

**REVISED**

P.12. Lines 22-27. Please fix several grammatical errors.

**REVISED**

P.14 Lines 9-13. This repeats what was already stated in

**REVISED**

P. 15. Lines 25-27: Another case of 2-sentence paragraph. Please combine with one of adjacent paragraphs

**REVISED**

P.17 Lines 11-12. The authors state that "There is significant degradation of validation accuracy if MODIS retrievals of Poor data quality (QA<3) are included in the analysis". This implies that QA values of 1 and 2 were used and that they gave "poor quality". I thought that only QA=3 and QA=0,3 were used. If I am correct, please change QA<3 to QA=0 to accurately describe the data used. Also please be consistent in the acronym for quality flag. You use QA in some places and QAF in others. Please pick one of them and use throughout the document.

**REVISED for clarity and we are now consistently using QAF throughout the manuscript.**

P. 17 Line 14. Please correct the dimensions of MODIS product. You state it as '3km2'. Please fix here and other instances in the paper.

**REVISED**

P 17 Line 23 – P. 18 Line 3: There are several grammatical errors in this paragraph (and similar errors in other sections of document), including missing commas and similar errors. Please fix throughout the document.

**REVISED for the grammar.**

P. 19. Lines 13-15. This is one of many sentences throughout the document which needs the grammar fixed. There are commas places where they should not be placed and missing from places where they should appear.

**REVISED for the grammar.**

**Response to doi:10.5194/amt-2018-44-SC1, 2018**
This Short Comment concerns the linear least-squares regression results presented in the
manuscript (in e.g. Table 1). I am posting it here after discussion with the authors in person.
While it is a commonly-used technique, unfortunately AOD data of this type are generally not
suitable for the use of ordinary least squares linear regression. The technique requires certain
assumptions about the nature of the data to be able to provide quantitatively meaningful
regression characteristics (and uncertainties on those characteristics), and these assumptions are
all questionable or violated in the case of remotely sensed AOD data of this type. For example,
assumptions of linearity, independence of data points, existence of a single population, Gaussian
behaviour of residuals, and scale-independence of AOD uncertainties. The result is that the
output numbers are not meaningful in the sense that we want to use them. It is not a matter of the
results being noisy; they can be systematically biased or in some cases meaningless.
I acknowledge that it is a commonly-used technique but that should not in my view be a valid
justification for doing something which is statistically inappropriate in a scientific journal. It is
best for us to stop doing it and in this way hopefully spread good practice more broadly through
the community.
The reason least-squares linear regression is a popular choice is it gives us two parameters
(intercept and slope) with which we can say something about what biases/offsets are in the
limiting cases of low-AOD and high-AOD regimes. The question then is what is the best way to
convey this type of information in a more statistically-appropriate way?
Fortunately the authors have largely already done so. Since we typically frame our retrieval
performance in terms of fraction within expected error (EE), the authors' inclusion of summaries
of what proportion of matchups are below, within, and above the EE is one welcome step.
Another is with the binned type of plots seen within e.g. Figure 6 (which incidentally already
shows that the relationships are overall not linear). The values of the offset for the low-AOD bins
provide an indication of typical biases in low-AOD conditions. And the relative magnitudes of
the offset for the high-AOD bins provide an indication of typical biases in high-AOD conditions.
Or if there is no apparent AOD-dependence then you can just state that the offset appears
invariant with AOD. I suggest that the authors remove least-squares slope and intercepts results
from the paper. For the same reason, ideally Pearson's linear correlation coefficient could also be
replaced with Spearman's rank correlation coefficient. If the authors wish to include replacement
information instead of slope/intercept to summarise the global statistics, I suggest adding
something like the magnitude and sign of absolute bias as seen in the low-AOD bins, and the
relative magnitude of the bias from the high-AOD bins.
For example, eyeballing from the bottom-left panel of Figure 6 (Terra, defined relative to
MODIS AOD), when MODIS retrieves AOD in the range -0.05 to 0 it looks like the typical
offset is about -0.05. When MODIS retrieves AOD above about 0.4, it looks like the bin
mean/median bias are positive and about 20%. So in this case you might say that the typical
biases are around -0.05 in the cleanest conditions and +20% in highAOD conditions. Or if you
take the top-left panel (Terra, defined relative to AERONET AOD), it looks like the bias it looks like the typical bias is around 0.05-0.1 regardless of AOD. In my view those numbers are more appropriate and more useful statistics to report than the regression slope/intercept.

**Thanks Dr. Sayer for posting your comments here and discussing with us in-person. These are important aspects of validation analysis.**

**As we discussed during the in-person meeting, we understand your concerns and we agree that AOD data may not follow all the assumptions required for an ideal regression analysis. In fact, we fail to find any suitable measurement in nature, which follows all these rules of regression strictly. Even so linear regression analysis has traditionally been and continues to be a useful tool for understanding, comparing with previous studies and especially in visualizing the relationship between two variables measured in nature.. If the relationship is not linear, seeing the cloud of points deviating from the drawn linear regression line is one of the most telling means to identify that non-linearity.  Seeing the linear regression line deviating from the one-to-one line is another simple, intuitive, first step in understanding the relationship between the variables. To be able to compare these relationships with similar exercises in previous studies, slope, intercept and correlation coefficients are provided. These standard parameters become the first set (but not the only set) of statistical parameters defining the performance of satellite retrieved AODs as compared to ground truth. Now, in order to further characterize the errors in satellite retrieved AODs, we provide additional statistics in the form of biases, expected errors and other useful parameters using standard statistical techniques.  We feel strongly that ALL analyses provided in the manuscript are of value in evaluating the satellite product, and we respectfully prefer to include linear regression in the paper.**

**We note that the rules and assumptions concerning linear regression analysis become more important when we intend to PREDICT a dependent variable with the help of an INDEPENDENT variable. For example, linear regression is insufficient when converting AOD into surface PM2.5. But, here in this study, we do not expect any reader to apply measured AERONET values of AOD to the calculated linear regression equations to predict MODIS values. Linear regression is a very poor model for such a purpose, but there is no practical reason why somebody would want to do so when AERONET makes much more accurate and precise measurements than MODIS.  Thus, the linear regression we present in this manuscript is an aid in understanding, not a statistical model for prediction, and for this reason we have decided to keep it in.**

[revised manuscript text omitted]

QAF=3. In the 10 km product there must be ≥ 51 native pixels surviving the selection process out of a possible 120 to reach this QAF level. The similar ratio for the 3 km product is ≥ 5 surviving pixels out of a possible 12. Any fewer, and there is insufficient statistical information for confidence in an aerosol retrieval. Therefore, for the 3 km retrieval, a situation with fewer than 5 native pixels automatically receives the designation of "poor quality" (QAF=0). For this resolution product there are no intermediate quality levels between 3 and 0 over land retrievals (Remer et al., 2013).

**3. Data Sets**

**3.1. MODIS 3 km AOD**

The primary data set of this study is the Collection 6 MODIS dark target retrieved aerosol optical depth at 3 km spatial resolution, derived from Terra reflectances (MOD04_3K), or Aqua reflectances (MYD04_3K), as described in Section 2. These are publicly available and can be downloaded from https://ladsweb.modaps.eosdis.nasa.gov/. Of the products in the data sets, we analyze only the AOD at 0.55 μm.

Applying identical algorithms to two different sensors does not guarantee identical results (Levy et al., 2015). The two MODIS DT data sets, one from MODIS-Terra and one from MODIS-Aqua must be addressed separately as individual and independent products, even though they have been created from identical algorithms with no specific tuning of parameters for each sensor. While MODIS-Terra and MODIS-Aqua began as near-identical sensors, they have evolved over their lifetimes to develop their own instrumental characteristics. For example, some detectors in Aqua's detector array at some wavelengths have died, resulting in fewer available reflectance pixels at those wavelengths. Terra's detector array has not lost any detectors. At the same time, we have seen drift in some of Terra's wavelengths, resulting in measureable artificial trends in the MODIS-Terra aerosol products (Levy et al., 2013; Sayer et al., 2015; Lyapustin et al., 2014). The most flagrant of those MODIS-Terra trends have been mitigated by aggressive radiometric calibration (Toller et al., 2013), which has been applied in creating the C006 DT products. Note that some projects (e.g. Lyapustin et al., 2014; Sayer et al., 2015) have since introduced additional calibration drift mitigation. However, the DT retrieval has not applied these strategies. In this work, we will analyze the C006 aerosol products from the two MODIS sensors, independently, to provide users with clear information on the strengths and limitations of each one.

**3.2. AERONET AOD**

The Aerosol Robotic Network (AERONET) is NASA's global ground network of CIMEL sun-sky radiometers that make measurement of directly transmitted solar light and scattered sky light at several wavelengths during daylight hours (Holben et al., 1998). In this work, only the direct sun measurements will be used. The AERONET group processes these spectral measurements to derive AOD at the wavelengths corresponding to the direct sun measurements. The AERONET spectral AOD product is a community standard for satellite-derived AOD validation, given that AERONET's AOD uncertainty of 0.01-0.02 (Eck et al., 1999) is sufficiently more accurate and precise than can be expected by any satellite retrieval. The typical temporal frequency of direct sun measurements is every 15 minutes. The network consists of hundreds of stations, located globally, across all continents and in a wide variety of aerosol, meteorological and surface type conditions. Only stations that sufficiently represent land areas will be used here, which means we are not comparing with observations taken on small islands, ocean platforms or mobile ships.

The configuration of the spectral bands varies, but typically is centered at 0.34, 0.38, 0.44, 0.50, 0.67, 0.87, and 1.02 μm. Here we use a quadratic log-log fit (Eck et al., 1999) to interpolate AERONET AOD to 0.55 μm to match the primary MODIS AOD product. AOD data from AERONET are reported for three different quality levels: unscreened (level 1.0), cloud screened (level 1.5) and cloud screened and quality assured (level 2.0). We will only use Version 2.0 Level 2.0 AERONET AODs in this study.

**4. Spatial and Temporal Collocation**

The validation procedure requires calculating the spatio-temporal statistics of a collocated MODIS-retrieved and AERONET-measured AOD pair (Ichoku et al. 2002; Petrenko et al., 2012;

Munchak et al., 2013; Remer et al., 2012). We have created a collocated data set (CDS) of AOD

at 0.55 μm from both MODIS-Terra and MODIS-Aqua, matched with AERONET, for nearly the entire mission (2003-2015 for Aqua and 2000-2015 for Terra). From here on, we use the term

"pixels" to refer to the MODIS retrieval product (e.g. 3 or 10 km resolution); if referring to the native MODIS pixel resolution (e.g. 0.5 km) we will denote as "native pixel".

In previous validation studies of the standard 10 km product the spatial statistics were based on groupings of either 5 x 5 MODIS product pixels (~50 x 50 km² box) centered on the AERONET

station (Ichoku et al., 2002; Levy et al., 2010) or all the MODIS product pixels within a 27.5 km radius around the AERONET station (Petrenko et al., 2012). These spatial statistics would be matched with the temporal statistics of ± 30 minutes of AERONET observations centered at satellite overpass time. These large spatial collocation boxes will not properly test the accuracy of finer resolution satellite products to represent small-scale aerosol gradients. Therefore, Remer et al., (2013) and Munchak et al. (2013) moved to a 7.5 km radius and ± 30 minutes satellite overpass.

The 7.5 km radius encompasses roughly 25 AOD pixels at nadir, which is analogous to the number of product pixels used with the coarser resolution product. In this study spatial statistics are calculated from all MODIS product pixels falling within a box of 0.15° x 0.15° (latitude x longitude) centered over an AERONET location. Except for Polar Regions, this is similar to a 15

x 15 km² box or 7.5 km radius at nadir. Temporal statistics are calculated from all AERONET

observations of AOD within ± 30 minutes of satellite overpass.

As recommended by the MODIS DT science team (Levy et al., 2010), unless otherwise specified, only AOD pixels with quality assurance flag 'very good' (QAF=3) are included in averaging over the AERONET sites. To be consistent with previous validation exercises (Levy et al., 2010), we have retained the collocated data sets only when there were at least 5 MODIS product pixels (out of a possible 25) and 2 AERONET measurements (out of a possible 2-4). The collocated data set (CDS) consists of 574 AERONET stations with 90,162 collocated pairs for MODIS-Terra and

71,248 collocated pairs for MODIS-Aqua. Figure 1 shows the locations of these stations and the color-coding represents the number of collocated AERONET-MODIS AOD pairs over the station.

Thus, a data set (i.e. CDS) of collocated MODIS-AERONET pairs of AOD at 0.55 μm is created that can be organized and subsampled in any number of configurations. In any subsample, or for the entire data set, these ordered pairs can be plotted, one against the other to create a scatterplot, and collocation statistics calculated.  We will use the following statistical parameters to quantify how well the MODIS retrievals match their collocated AERONET counterparts (Hyer et al, 2011):

• Correlation coefficient (R),

• Slope of the linear regression line,

• Root Mean Square Error (RMSE)

$$\text{Mean Bias} = \frac{1}{N}\sum(\text{MODIS AOD} - \text{AERONET AOD}) \cdots\cdots\cdots\cdots (1)$$

Percentage of collocations falling within expected error,

$$\text{EE} = \pm(0.05 + 0.20 \times \text{AERONET AOD}) \cdots\cdots\cdots\cdots\cdots\cdots (2)$$

Error Ratio (ER),

$$ER = (\text{MODIS AOD} - \text{AERONET AOD})/\text{EE} \cdots\cdots\cdots\cdots\cdots\cdots (3)$$

The coefficients in the EE equation were determined from evaluation of the 3 km product over the six months of Aqua data analyzed by Remer et al., (2013). Those limited results suggested that expected error bounds should be broadened to the values seen in Eq. (2) from those derived for the

10 km product (EE=±[0.05+0.15 x AERONET AOD]).

The number of collocations (N) is another parameter used to evaluate the 3 km retrieval in the collocation data set.

**5.  Validation Results**

**5.1. Global Statistics**

We first compare MODIS 3 km AOD retrievals against collocated AERONET values (Figure 2), for both the recommended 'high quality' retrievals (QAF=3) and for all the retrievals, regardless of quality, keeping Terra and Aqua results separate. Note that the 3 km product only tags data as either 'high quality' or 'low quality'. Table 1 presents the statistical parameters corresponding to
this analysis while considering various combinations of QAFs.
Globally, there is strong correlation between MODIS 3 km AOD and collocated AERONET
equivalents. However, there is scatter and a positive bias to the retrievals, more so for Terra than
Aqua, even though the correlation is similar between the satellites. The retrieval quality identified
by the algorithm corresponds well to the product accuracy as determined by collocation with
AERONET observations. Algorithm-identified high quality retrievals (QAF=3) have stronger
correlation, smaller RMSE and more retrievals falling within expected error than do the low quality
(QAF=0) retrievals (Table 1). However, the high quality data set contains about 20% fewer
retrievals than does the total data set with retrievals of all quality levels included. Figure 3 shows
that the differences between Terra and Aqua in how they match AERONET values are much more
apparent than the differences between QAF levels of the same satellite sensors. We note that only
the high quality (QAF=3) Aqua 3 km retrievals meet expectations in terms of falling within the
standard expected error bars (Remer et al., 2012; and Eq. 2).
Table 1 also shows the corresponding validation statistics for the 10 km product for QAF=3,
distinguishing between Terra and Aqua. The 10 km product, as expected, more closely matches
AERONET values, having higher correlation, lower bias and RMSE, and producing more
retrievals that fall within expected error bounds than does the 3 km product. We note that even in
the 10 km validation statistics, mean bias for Terra is 0.03 higher than for Aqua, which is the same
difference between sensors as found for the 3 km product. The results in Table 1 confirm Remer
et al. (2013)'s conclusion that the 3 km product is less accurate than the standard 10 km product.
The remainder of the paper will be devoted to exclusively analyzing the differences between the 3
km product and AERONET, without further reference to the standard 10 km product.

**5.2. Regional Statistics**
The accuracy of the 3 km AOD retrievals will be regionally and locally specific, depending on
how well retrieval assumptions of surface and aerosol optical properties match actual conditions.
Local cloud conditions also may introduce uncertainty into the retrieval. Furthermore, the spatial/temporal variability of the area may create biases in the collocation methodology that depends on assumptions of aerosol homogeneity.  Here we investigate how well the MODIS 3 km product matches AERONET over individual AERONET stations.

For the regional and local analyses, we will use only QAF=3 retrievals and calculate the same collocation statistics for each station individually. Figure 4 plots the values for correlation coefficient, mean bias, percentage within expected error, and RMSE for each station that reported at least 100 collocations over the entire time series.  In general, the MODIS 3 km retrievals show high correlations over much of the northern mid-latitudes where there are AERONET stations in abundance. Correlation is weaker at some stations in California and the arid southwest of North

America, in the Caribbean, Central America, Insular SE Asia, Australia, and especially in southern

South America.  These are locations where the standard 10 km product also shows poor agreement with AERONET (https://darktarget.gsfc.nasa.gov/validation/maps).  In most of these regions, like the arid southwest of North America, the surface properties do not agree with the assumptions used in the global retrieval, thereby introducing error in the retrieval.

Not all stations with strong correlations exhibit small mean biases.  For example, MODIS 3 km retrievals severely under predict AOD in the stations of west Africa, falling well below expected error there, even though those stations report high correlations with AERONET.  Such a validation pattern is symptomatic of incorrect assumptions of aerosol properties. In west Africa, the interplay of heavy dust and heavy smoke, often occurring simultaneously in the atmospheric column at the same time, creates difficult situations to properly model in the aerosol retrieval.  Likely the poor agreement between MODIS and AERONET there can be attributed to this difficulty.  Stations in

Australia, show relatively small mean biases and high percentages meeting expectations, despite poor correlations. This apparent contradiction suggests that the poor correlations are the result of small dynamic range in the scatter plots that occur when AOD is consistently low.

In Figure 4, we see the local nature of the validation statistics.  Stations in close proximity to each other sometimes report very different statistics.  For example, the stations clustered across northern

India, and those in an array across central South America (Brazil) range from strong positive to negative mean biases and RMSE error from 0.05 to 0.20, even though these groupings of stations will fall within the same region as defined in Figure 5.  This is apparent in almost any region.  Some of this variability may be due to differences in the temporal extent of the AERONET record at each individual station, so that even if stations are in close proximity in space, they may actually be making measurements in entirely different years or seasons.  Other differences may be related to topography, urban surfaces, or other factors. Still, the variability seen in Figure 4 shows how local conditions, and possibly the individual characteristics of the time series affect validation statistics.

The final point to note in Figure 4 is the difference between Terra and Aqua.  For example, in the mean bias plots we see how the mean bias across the North American central plains decreases from approximately positive 0.04-0.05 for Terra to slightly negative for Aqua.  For many of the stations, positive mean AOD biases are higher for Terra than for Aqua.  This is in agreement with the global statistics presented in Table 1.

We next group individual stations into 17 regions, defined following Hyer et al., (2011). These are shown in Figure 5, with Table 2 presenting the regional validation statistics for each of the defined regions.  We know from previous analyses presented above that there are distinctive differences between Terra and Aqua mean biases; however, in calculating the regional statistics of Table 2, we combine Terra and Aqua collocations.

[revised manuscript text omitted]

This validation study only addressed the 3 km AOD product over land, and did not evaluate the over water product. The study took a global and regional view, not a local one. Users of the product on a local level are encouraged to consider particular biases that may occur due to local conditions. For example, we know that the MODIS Collection 6 dark target AOD retrieval is systematically biased over urban surfaces (Gupta et al., 2016). This is true for both the 10 km and 3 km dark target products. This problem has been addressed and is substantially mitigated with the release of the Collection 6.1 version of the algorithm (Gupta et al., 2016). In the meantime, the results here show that overall the dark target MODIS Collection 6 algorithm is producing an AOD product at 3 km resolution with sufficient accuracy and with biases well-characterized. The product can now be used quantitatively in a wide variety of science and practical applications.

**7. Acknowledgement**

This work was supported by the NASA ROSES program NNH13ZDA001N-TERAQEA: Terra and Aqua – Algorithms – Existing Data Products and NASA's EOS program managed by Hal Maring. We thank MCST for their efforts to maintain and improve the radiometric quality of MODIS data, and LAADS/MODAPS for the continued processing of the MODIS products. The AERONET team (GSFC and site PIs) are thanked for the creation and continued stewardship of the sun photometer data record; which is available from http://aeronet.gsfc.nasa.gov.

[revised manuscript text omitted]